

**Impact of skin effect on single-well push-pull tests with the presence of regional**
**groundwater flow**
Xu Li[a], Zhang Wen[a]*, Hongbin Zhan[a,b], Qi Zhu[a]
[a]School of Environmental Studies, China University of Geosciences, Wuhan, 430074, China
[b]Department of Geology and Geophysics, Texas A & M University, College Station, TX
77843-3115, USA, Email: zhan@geo.tamu.edu
*Corresponding author: Zhang Wen, Ph.D. Professor. Affiliation: School of Environmental
Studies, China University of Geosciences, Wuhan, Hubei, 430074, P. R. China. Email:
wenz@cug.edu.cn. Tel: 86-27-67883159. Fax: 86-27-87436235.





**Abstract**
Single-well push-pull (SWPP) test is one of the most important ways to estimate aquifer
transport parameters, e.g. porosity, dispersivity, rate of biogeochemical reaction, but its
application for determining the regional groundwater velocity has rarely been discussed in
previous studies. In this study, a new numerical model of SWPP test considering regional
groundwater flow and skin effects was established using the finite-element COMSOL
Multiphysics. The effects of regional groundwater flow velocity and skin properties on
breakthrough curves (BTCs) were thoroughly analyzed. Several important results were
obtained. Firstly, the regional groundwater velocity affects the types of BTCs through
changing the pattern and location of the dividing streamline. Secondly, a positive (or
negative) skin leads to a slower (or faster) tracer transport process. That is, a positive skin
results in a higher concentration at early stage at a given time. Thirdly, a smaller hydraulic
conductivity ratio $\delta$ of the positive skin to the formation results in greater solute plume
retardation in the skin zone. Besides, a larger thickness of the positive skin leads to a higher
tracer concentration around the well. The opposite is true if the skin is negative. The general
conclusion is that the skin effects on SWPP test are significant and should be considered.
**Keywords: push-pull test, regional groundwater velocity, solute transport, skin effect**



## 1. Introduction

The single-well push-pull (SWPP) tests have been commonly employed to estimate
aquifer parameters, e.g. regional groundwater flow velocity, porosity, dispersivity,
biogeochemical reaction rate (Gelhar and Collins, 1971; Hall et al., 1991; Schroth and Istok,
2006). The process of this test can be summarized as follows: A tracer is injected into a target
aquifer (push), then the mixed solution is pumped out from the same location (pull).
Groundwater samples are taken at regular time intervals at the test well during the pumping
process, and parameters can be obtained by fitting the observed breakthrough curves (BTCs)
using a proper mathematical model. Generally, a complete SWPP test may consist of four
phases: tracer injection, chasing, rest, and pumping. The chasing phase is to push the tracer
away from the injection well (Istok et al., 1997), and the rest phase is for tracer to diffuse
and/or react with the aquifer (if a reactive tracer is employed). Conservative or reactive
tracers can be utilized, depending on the purpose of the test. In general, conservative tracers
have been widely used to estimate regional groundwater flow velocity, porosity, and
dispersivity, etc. (Leap and Kaplan, 1988; Haggerty et al., 2001; Hebig-Schubert, 2014).
Similarly, one can obtain the information of sorption, cation exchange, microbial processes
by applying reactive tracers (Trudell et al., 1986; Field et al., 2000; Tong et al., 2016). For
instance, Tong et al. (2016) used SWPP tests to validate the abundant production of hydroxyl
radicals due to the oxidization of subsurface sediments.
To interpret the SWPP test results, a proper mathematical model considering the
fundamental physical and biogeochemical processes of the test is indispensable (Haggerty et
al., 2001; Kleikemper et al., 2002; Schroth and Istok, 2006). From a transport perspective,



many existing models are governed by the conventional advection-dispersion equation
(ADE), assuming the validity of Fick's law in the SWPP tests (Schroth et al., 2000; Huang et
al., 2010). Subsequently, many analytical and numerical solutions of various single-well
models have been developed. For instance, Huang et al. (2010) obtained an exact analytical
solution of SWPP test by using Fick's law, considering a partially penetrating well in the
aquifer. Besides ADE, a number of non-Fickian transport models of SWPP tests have also
been developed in recent years to recognize the influence of media heterogeneity, especially
in fractured aquifers (Chen et al., 2017). Such models include multi-rate mass transfer models
(Haggerty et al., 2001), continuous time random walk (CTRW) (Le Borgne and Gouze,
2008), and fractional advection-dispersion equation (FADE) models (Benson et al., 2004;
Chen et al., 2017), to name a few. For instance, Chen et al. (2017) developed a fractional
model of multistage SWPP test to simulate non-Fickian behavior for a fractured aquifer. In
addition, Schroth et al. (2005) obtained an approximate analytical solution of SWPP test for
spherical-flow conditions. Wang et al. (2017) investigated the impacts of transient flow and
wellbore storage on SWPP test under transient Forchheimer flow using a finite-difference
method.
As mentioned above, SWPP test is a powerful tool for aquifer characterization, including
the determination of regional groundwater flow velocity. Traditionally, regional groundwater
flow velocity can be obtained by three or more groundwater monitoring wells in the aquifer
by conducting the natural gradient tracer tests rather than the SWPP tests (Pickens et al.,
1981; Michie, 1996; Zimmermann and Huenges, 1999). However, the natural gradient tracer
tests usually take much longer time to complete. This is particularly troublesome when the



medium is less permeable, and the regional groundwater flow velocity is relatively small
(Schubert et al., 2011). The natural gradient tracer test method is also very costly to
implement in deep aquifers as the installation of multiple wells in deep aquifers can be
formidably expensive. In contrast, the SWPP test only needs a single well, thus can
substantially reduce the cost of test, and becomes a nice alternative for the determination of
regional groundwater flow velocity (Leap and Kaplan, 1988; Butler et al., 2009). To serve
such a purpose, the SWPP tests usually consist of three phases: tracer injection, rest, and
pumping. The rest phase allows the injected tracer to drift with regional groundwater
velocity, thus it is a key phase to include. Leap and Kaplan (1988) obtained an equation for
push-pull test to determine regional groundwater flow velocity in a confined aquifer, and a
series of laboratory tests were conducted to verify the accuracy of the model, but the test
results showed that if the solute transport drifted over the location of dividing streamline
toward downstream, the calculated results of the regional flow velocity would produce a large
error with the comparison to the actual value. After that, Hall et al. (1991) presented another
type of SWPP test for determining regional groundwater velocity and effective porosity based
on the method of Leap and Kaplan (1988), but Hall et al. (1991) required a directly
downgradient monitoring well. It is notable that both approaches mentioned above have some
limitations for determining regional groundwater velocity.

In addition, the impacts of skin near a pumping well are usually neglected for a SWPP

test, which might bring about great errors for the estimation of aquifer parameters and
regional groundwater flow velocity. During the process of well implementation, the intrusion
of drilling mud into the aquifer in the vicinity of well is inevitable, which can result in the





change of the porosity and permeability surrounding the well screen (Hurst et al., 1969). This
phenomenon can be regarded as the skin effect (Chen and Chang, 2002; Wang et al., 2012).
The thickness of skin usually ranges from a few millimeters to several meters (Novakowski,
1989). The skins can be classified into positive and negative types according to the hydraulic
conductivity contrast between the skin and the formation zone. If the hydraulic conductivity
of the skin is smaller than that of the formation zone, the skin is defined as a positive one.
Otherwise, it is a negative skin (Park and Zhan, 2002; Yeh et al., 2003; Wen et al., 2011).
Such a skin, regardless of positive or negative, will inevitably alter the flow field near the test
well, thus its effect must be taken into consideration for interpreting the SWPP test. For
instance, the streamlines of skin zone can converge toward the well in the case of a negative
skin, but the opposite is true for a positive skin (Drost et al., 1968; Schubert et al., 2011).

In summary, the skin effect is a very important issue from the perspective of SWPP test

interpretation. Through a careful check on the literature, we notice that the model of SWPP
for estimating groundwater flow velocity needs further investigation, besides, the impacts of
skin effects on SWPP tests for estimating groundwater flow velocity have rarely been
studied, which will be the purpose of this study. To accomplish the objective, we will
investigate a SWPP test containing three phases of injection, rest and pumping using a fully
penetrating well. We will use the finite-element COMSOL Multiphysics to numerically
simulate the steady-state, two-dimensional (2D) horizontal flow, with specific attention paid
to the skin effect.
**2. Mathematical model of the SWPP test**





To illustrate the problem, we will use a conservative tracer. A confined aquifer is
assumed to be unbounded laterally with a uniform regional groundwater flow presented over
the entire duration of test. The aquifer is assumed to be homogeneous and horizontally
isotropic. A fully penetrating well is used so only the horizontal flow is of concern here. Flow
is assumed to be Darcian and transport is assumed to be Fickian. The test well radius is
assumed to be sufficiently small so that the wellbore effect is not a concern. The tracer is
injected with a constant rate and a constant concentration at the injection phase and is
pumped with a constant rate (which could be different from the injection rate) at the pumping
phase, after a certain period of rest phase to allow the injected tracer drifting with the regional
groundwater flow. The coordinate system is established as follows with the origin at the
center of the test well and the *x*-axis pointing to the direction of regional groundwater flow. A
2D schematic diagram investigated here is depicted in Fig.1.
**2.1 Mathematical model of groundwater flow**
Flow of the SWPP test is assumed to be steady-state, thus the groundwater flow velocity
can be expressed as the superposition of the flow component generated by the pumping well
and the regional flow:
$\vec{v} = \vec{v}_1 + \vec{v}_2$ (1)
$\vec{v}_1 = v_1 \vec{e}_r = Q / (2\pi\theta B r) \vec{e}_r$ (2)
$\vec{v}_2 = v_2 \vec{e}_x = (-KJ / \theta) \vec{e}_x = (v_d / \theta) \vec{e}_x$ (3)
$r = \sqrt{x^2 + y^2}$ (4)
where the arrow over a symbol represents a vector hereinafter; $\vec{v}_1$ is the average radial pore
velocity vector generated by the injection (or pumping well) with a magnitude of $v_1$ [L/T]and





$\vec{e}_r$ is a unit vector along the radial direction; $\vec{v}_2$ is the regional groundwater pore flow
velocity vector with a magnitude of $v_2$ [L/T] and $v_d$ is the regional groundwater Darcy flow
velocity, and $\vec{e}_x$ is a unit vector long the $x$-axis; $\vec{v}$ is the lumped groundwater flow velocity
near the well [L/T]; $B$ is the aquifer thickness [L]; $r$ is the radial distance [L]; $Q$ is the
injection or pumping rate [L$^3$/T], which is positive for the injection and negative for pumping,
and $Q$ is 0 during the rest phase; $K$ is aquifer hydraulic conductivity [L/T]; $J$ is the hydraulic
gradient of regional flow [L/L]; $\theta$ is the aquifer effective porosity [dimensionless], which is
assumed to be the same as the total porosity of the aquifer when all the pore spaces are well-
connected with negligible immobile porosity; $r$ is the radial distance [L] from the well and $x$
and $y$ are two horizontal coordinates [L], parallel and perpendicular to the regional
groundwater flow direction, respectively.
According to Fig.1, the boundary conditions for the domain of concern can be expressed
as:
$H(x,y)\big|_{s_1} = H_1, \quad H(x,y)\big|_{s_2} = H_2$                       (5)
$K\dfrac{\partial H}{\partial n}\bigg|_{s_3} = 0, \quad K\dfrac{\partial H}{\partial n}\bigg|_{s_4} = 0$                 (6)
where $s_1$, $s_2$, $s_3$ and $s_4$ are the boundaries of the model; $s_1$ and $s_2$ are constant-head boundaries
with prescribed heads of $H_1$ and $H_2$, respectively; both $s_3$ and $s_4$ are no-flux boundaries.
Therefore, a constant regional flow field can be generated and one can obtain different values
of $v_2$ by changing the head differences of $H_1$ and $H_2$.
**2.2 Mathematical model of solute transport**
The ADE of a conservative solute without source/sink can be written as:





$$\frac{\partial C}{\partial t} = \nabla \cdot (D \nabla C) - \nabla \cdot (\vec{v} C)$$ (7)
where $C$ is the solute concentration [M/L$^3$]; $t$ is the transport time [T]; $D$ is the hydrodynamic
dispersion [L$^2$/T]; $\nabla \cdot$ and $\nabla$ are the divergence operator and the gradient operator
respectively; the hydrodynamic dispersion is a velocity-dependent tensor depicted as:
$$D_{xx} = \frac{\alpha_L v_x^2}{|\vec{v}|} + \frac{\alpha_T v_y^2}{|\vec{v}|} + D_{diff}$$ (8)
$$D_{yy} = \frac{\alpha_L v_y^2}{|\vec{v}|} + \frac{\alpha_T v_x^2}{|\vec{v}|} + D_{diff}$$ (9)
$$D_{xy} = D_{yx} = (\alpha_L - \alpha_T) \frac{v_x v_y}{|\vec{v}|}$$ (10)
$$v = \sqrt{v_x^2 + v_y^2}$$ (11)
where $D_{xx}$, $D_{xy}$, and $D_{yy}$ are components of the hydrodynamic dispersion coefficient tensor
[L$^2$/T]; $D_{diff}$ is the molecular diffusion coefficient [L$^2$/T]; $\alpha_L$ is the longitudinal dispersivity
[L]; $\alpha_T$ is the transverse dispersivity [L]; the transverse dispersion effect is much smaller, thus
one usually assumes $\alpha_L = 10\alpha_T$ (Guvanasen and Guvanasen, 1987; Chen et al., 1999; Chen et
al., 2006); $v_x$ is pore velocity in the $x$ direction; $v_y$ is pore velocity in the $y$ direction.
The initial condition is:
$$C(r,0) = 0,$$ (12)
During the injection phase, the inner boundary condition inside the well can be
described as:
$$C(r,t) = C_0, \qquad r = r_w, 0 < t < t_{inj}$$ (13)
where $C_0$ represents the concentration of the injection phase [M/L$^3$]; $r_w$ is the well radius [L];
$t_{inj}$ is the duration of the injection phase [T]. The third-type boundary condition may also be



used to replace the first-type boundary condition of Eq. (13). However, our numerical
exercises indicate that both conditions yield nearly the same results except for a very short
period of time since the start of injection. Therefore, without loss of generality, we use the
first-type boundary condition here as an example to illustrate the methodology.
During the rest phase, the solute flux from the borehole into the aquifer is zero, but the
solute concentration around the borehole is nonzero, therefore, rather than a constant
concentration boundary, a constant-flux (or the third-type) boundary is more reasonable and
can be described as:
$$v_1 C - D \frac{\partial C}{\partial r}\bigg|_{r \to r_w^-} = 0, \qquad t_{inj} < t < t_{res} \qquad\qquad (14)$$

where $t_{res}$ is the duration of the rest phase [T]. Eq. (14) represents a zero-flux boundary.
During the rest phase, $Q$ is 0.
During the pumping phase, the time-dependent concentration is measured in the
borehole. The main target of the test is to obtain several parameters by fitting the observed
breakthrough curves (BTCs) with corresponding theoretical BTCs obtained from a proper
analytical or numerical solution. When solute transport through well screen, the boundary
condition at the well screen is (Wang et al., 2017):
$$\frac{\partial C}{\partial r}\bigg|_{r \to r_w^-} = 0 \qquad\qquad (15)$$

Because the values of velocity and concentration are different around the perimeter of
the borehole, it is necessary to integrate the concentration around the borehole with the
velocity as the weight to obtain the accurate value of concentration at the well, thus, the flux-
averaged concentrations can be expressed as:



$$\overline{C} = \frac{\oint_{l_w} v_w C_w}{\oint_{l_w} v_w}, \qquad r = r_w, t_{res} < t < t_{pump} \tag{16}$$
where $v_w$ represents the superposition velocity around the borehole during the pumping [L/T];
$C_w$ represents the concentration around the well perimeter [M/L3]; $\overline{C}$ is the concentration
inside the well [M/L3]; $l_w$ is the perimeter of the wellbore [L].
**3. Numerical solution of the SWPP test**
In this study, a steady-state flow model of 2D horizontal plane was developed based on
COMSOL Multiphysics, as shown in Fig. 1. The model region was set to be 40 m $\times$ 40 m,
and the well has a radius of 0.1 m. In addition, $B$=10 m, $K$=8.0 m/d, $\theta$=0.3. In this model,
constant-head boundaries were prescribed, and the value of $H_2$ was set as constant 15 m,
according to Eq. (3), one can obtain different values of $v_2$ by changing the value of $H_1$. In the
model, a continuous mass flux of injection or pumping rate was assigned at $r=r_w$, which can
be expressed as:
$$N_0 = \frac{Q\rho}{2\pi r_w B} \tag{17}$$
where, $N_0$ is the mass flux per unit thickness [M/L$^2$/T]; $\rho$ is the density of groundwater
[M/L$^3$]. The SWPP test was divided into three phases, the simulation results at the end of
each phase, including the hydraulic head and the solute concentration, were set as the initial
values for the simulation in the next phase.
A uniform skin near the well was considered in a confined aquifer, and the thickness of
the well skin was assumed to be constant and equal to $r_s$ along the well screen in this model.
The skin hydraulic conductivity and effective porosity were set as $K_s$ and $\theta_s$ respectively. The
default values of the parameters were shown in Table 1.





The model domain was discretized into 21688 elements, and the mesh size was
progressively refined near the well. When the number of element is doubled, the peak solute
concentration for the pumping phase varied about 0.17%. Therefore the selected mesh is
regarded as sufficiently fine for the problem investigated here. To check the accuracy of the
numerical model further, the numerical solution for a special case (without skin) was used to
compare with the analytical solution of Huang et al. (2010), who investigated a steady-state
flow SWPP model with injection and extraction phases, without the regional groundwater
flow, as shown in Fig.2. The simulated time span of tracer injection and pumping were 0.5
and 1 day, respectively. The other parameters were given as: $Q_{inj}$= 50 m$^3$/d, $Q_{pump}$=-50 m$^3$/d,
$B$=10 m, $\alpha_L$=0.1 m, 0.5 m, 1 m, and $\theta$=0.3. $C_0$ at $r=r_w$ was set as 1.0 mol/m$^3$. The results
showed that our numerical solution agreed perfectly with the analytical solution. For the
following analysis, the default values of the parameters are listed in Table 1.
**4. Results and discussions**
**4.1. Effects of regional groundwater velocity on BTCs in the SWPP test**
Fig. 3 shows the BTCs for the pumping phase with different regional groundwater
velocities, like $5\times10^{-7}$ m/s, $1\times10^{-6}$ m/s, $1.5\times10^{-6}$ m/s, $2\times10^{-6}$ m/s, $2.5\times10^{-6}$ and $3\times10^{-6}$
m/s. Besides, $\alpha_L$ =0.1 m, $Q_{inj}$= 30 m$^3$/d, $Q_{pump}$=-15 m$^3$/d, and the other parameters are the
same as those used in Table 1. It is found that different regional groundwater velocities have
great impacts on BTCs, and such impacts depend on the value of the regional groundwater
velocities. It is found that the tracer concentration is smaller at early stage with a greater
regional groundwater velocity. Additionally, it is notable that a larger regional groundwater
velocity will result in a longer tailing.





Fig. 4 shows the superposition of flow components generated by the pumping well and
the regional flow. For the pumping phase, one can see that there is a stagnation point (Sp)
located at the dividing streamline (Ds) as shown in Fig.4. For a single tracer particle, whether
it can be extracted out from the aquifer depends on the pattern and location of the dividing
line, which acts like a fishing net to collect all the products together. Therefore, different
convergence situations of tracer due to variable dividing streamlines result in a series of BTC
types. In order to interpret this behavior explicitly, the concentration distributions in a 2D
horizontal plane at $t_{pump}$=0 hr with different regional groundwater velocities are shown in Fig.
5. One can see that a certain amount of tracer mass may be retained near the symmetry axis
($x$-axis) and around the well when the regional groundwater velocity is relatively low, such as
$v_d$=1×10⁻⁶ m/s and 1.5×10⁻⁶ m/s, as shown in Figs.5a-5b, resulting in relatively high
concentrations in the wellbore at early stage. On the contrary, a larger regional groundwater
velocity leads to a faster tracer transport process, causing the tracer mass drifting away from
the well, as shown in Figs.5c-5d. Besides, one can also see that a larger regional groundwater
velocity leads to a smaller distance from Sp to well, resulting in a smaller portion of tracer
mass that can be extracted during the pumping phase. The opposite is true for the case of a
smaller regional groundwater velocity. For instance, as for the tracer mass on the left side of
the dividing streamline, they can be extracted by smaller velocities such as $v_d$=5×10⁻⁷ m/s
and 1×10⁻⁶ m/s, as shown in Figs.5a-5b, but very limited tracer can be captured with larger
velocities like $v_d$=3×10⁻⁶ m/s. And this further confirms the reasonability of the BTC types in
Fig. 3.
**4.2 The effects of $t_{res}$ on BTCs in the SWPP test**





Fig. 6 shows BTCs for the pumping phase with different $t_{res}$, like 6, 12, 24, and 36 hr.
Besides, $\alpha_L$ =0.1 m, $v_d$ =$3\times10^{-6}$ m/s, $Q_{inj}$=30 m$^3$/d, $Q_{pump}$=-15 m$^3$/d and the other parameters
are the same as those used in Table 1. It is found that the tracer concentration is smaller as $t_{res}$
increases in Fig. 6. This is because a longer time of rest phase means a farther distance of
tracer drifting, leading to a smaller portion of tracer mass that can be extracted during the
pumping phase. In order to interpret this behavior explicitly, the concentration distributions in
a 2D horizontal plane at $t_{pump}$=0 hr with different $t_{res}$ are shown in Fig. 7. One can see that a
longer $t_{res}$ means that more tracer mass drifting over the location of Sp toward downstream,
resulting in lower concentrations in the wellbore during the pumping phase in Fig. 7.
According to the analysis above, there is a strong interaction between regional groundwater
flow and well flow, thus proper choices of the duration of each phase, and the injection and
pumping rates are vital for the success of a SWPP test. For instance, for the case of a
relatively large regional groundwater velocity, one can decrease $t_{res}$ or increase the magnitude
of $Q_{pump}$ to recollect the tracer as much as possible, thus avoiding the over- or under-
estimation of hydraulic parameters from the SWPP test.
**4.3. The effects of porosity on BTCs in the SWPP test**
Fig. 8 shows the effects of porosity $\theta$ on BTCs during the pumping phase. The
parameters are given as: $\theta$= 0.1, 0.2, 0.3, 0.4 and 0.5 respectively, $v_d$=$3\times10^{-6}$ m/s, $\alpha_L$=0.1m,
$Q_{inj}$=30 m$^3$/d, and $Q_{pump}$=-30 m$^3$/d. It can be found that the concentration is smaller at early
stage with a smaller porosity. It is also obvious that a smaller $\theta$ will result in a longer tailing
at late stage. The explanation is similar to that for Fig. 3, i.e., a smaller $\theta$ means a faster pore
velocity, resulting in faster solute transport according to Eq. (3).



**4.4. The effects of dispersivity on BTCs in the SWPP test**


Fig. 9 shows the effect of $\alpha_L$ on BTCs for the pumping phase. The values of $\alpha_L$ are set as:
$\alpha_L$ =0.01 m, 0.05 m, 0.1 m, 0.5 m and the other parameters are given as: $v_d$=2×10⁻⁶ m/s,
$Q_{inj}$=30 m³/d, and $Q_{pump}$=-30 m³/d. As shown in Fig. 9, one can see that $\alpha_L$ has a significant
impact on BTCs. At early stage of pumping, the concentration shows a decreasing trend with
increase of $\alpha_L$ , This is because a larger dispersivity means a faster tracer transport , given the
same regional groundwater velocity, which causes much broader solute plume after the
injection and rest phases. A smaller dispersivity means a narrower solute plume. Therefore,
different dispersivities can change the characteristics of BTCs under the influence of regional
groundwater velocity.

**4.5. The effects of skin hydraulic conductivity on BTCs in the SWPP test**


As mentioned above, the well skin includes two general types, i.e., a positive skin or a
negative skin. Denoting the hydraulic conductivities of skin and aquifer (or formation zone)
respectively as $K_s$ and $K$, one can use a new parameter $\delta=K_s/K$ to reflect the skin impact,
where $\delta$ is a parameter reflecting the type of the skin and called the "skin index" hereinafter.
Specifically, $\delta<1$ represents a positive skin, while $\delta >1$ represents a negative skin. Note that
the case of $\delta$ =1 represents the case without a skin. In this section, we will provide a detailed
analysis on the impact of the skin hydraulic conductivity on SWPP test.

**4.5.1 Positive skin**


Fig. 10 shows the effect of different skin indexes on BTCs for a positive skin case
during the pumping phase. The parameters are given as: $v_d$=3×10⁻⁶ m/s, $Q_{inj}$=30 m³/d,
$Q_{pump}$=-30 m³/d, $\alpha_L$=0.1 m, $r_s$=0.6 m, $\delta$= 1, 0.5, 0.25 and 0.125, respectively. The results





indicate that the concentration gets higher at early stage of pumping when the skin index is
lower, as shown in Fig. 10. This may be explained as follows. A skin with a lower $\delta$ value (or
a lower permeability value in respect to that of the formation) essentially serves as a
somewhat "shield" around the test well that can make the spreading of the tracer mass out of
the test well more difficult during the injection phase. Consequently, more tracer mass will be
retained near the test well either in the skin or near the skin in the formation zone. Therefore,
during the pumping phase of the test, more tracer mass can be extracted during the early stage
of the pumping phase, leading to higher concentration during that stage. To further explicitly
interpret this behavior, the concentration distributions in a 2D horizontal plane at $t_{pump}$=0 hr
with different $\delta$ values are shown in Fig. 11. As can be seen in Fig.11, a lower skin hydraulic
conductivity leads to more tracer accumulation in the skin zone after the rest phase. For
instance, one can see that the solute plumes with high concentrations are clearly visible in
skin zones for the cases of $\delta$=0.25 and 0.125.
**4.5.2 Negative skin**
Fig. 12 shows the effect of different skin indexes on the BTCs for a negative skin case.
The parameters are given as: $v_d$=3×10$^{-6}$ m/s, $Q_{inj}$= 30 m$^3$/d, $Q_{ext}$=-30 m$^3$/d, $\alpha_L$=0.1 m, $r_s$=0.6
m, and $\delta$=1, 1.5, 2 and 3 . In contract to what has been observed in Fig. 10 for a positive skin,
the results indicate that the concentration gets lower at early stage of pumping when the skin
index ($\delta$) increases. This is because a negative skin is somewhat like a "high conductance
zone" rather than a "shield", and can facilitate the spreading of tracer mass away from the test
well during the injection phase. Therefore, less tracer mass will be retained near the test well
for a higher skin index, thus less concentration will be seen during the early stage of pumping




in the wellbore. Similar to what has been done for a positive skin in Fig. 11, the concentration
distributions in a 2D horizontal plane at $t_{pump}$=0 hr with different skin indexes are shown in
Fig. 13. It is quite obvious to see that a larger skin hydraulic conductivity leads to a less tracer
concentration in the skin zone at the early stage of the pumping phase. For instance, one can
see that tracer accumulation in the skin zone for the cases of $\delta$=2 and 3 are clearly less than
the cases of $\delta$=1 and 1.5.
**4.6. Effects of skin thickness on BTCs in the SWPP test**

In this section, we will analyze the impacts of the skin thickness on BTCs.

**4.6.1 Positive skin**

Firstly, we will analyze the impacts of the skin thickness on BTCs for a positive skin

case. The parameters are given as: $v_d$=3$\times$10$^{-6}$ m/s, $Q_{inj}$= 30 m$^3$/d, $Q_{ext}$=-30 m$^3$/d, $\delta$= 0.5
(positive skin), $\alpha_L$=0.1 m, and $r_s$= 0, 0.2, 0.4 and 0.6 m, respectively. Fig. 14 shows the
effects of the skin thickness (positive skin) on BTCs during the pumping phase. One can see
that the concentration gets higher at early stage with the increase of $r_s$. The explanation is
similar to that for Fig. 10, as a thicker positive skin means a thicker "shield" surrounding the
test well, preventing the injected tracer mass from spreading further away from the test well,
thus leading to higher concentrations during the early stage of the extraction phase. The
concentration distributions in a 2D horizontal plane at $t_{pump}$=0 hr with different positive skin
thickness are shown in Fig. 15. It is evident that the tracer mass still accumulates in the skin
zone 24 hr after the cease of the injection phase (rest phase), and the concentration is higher
in the skin zone than that in the formation zone (see the case of $r_s$=0.6 m). Besides, more



tracer can be found in the skin region with the increase of $r_s$, resulting in different shapes of
BTCs in Fig.14.

**4.6.2 Negative skin**

Similarly, we have also analyzed the impacts of the skin thickness on BTCs for a
negative skin case. The parameters are given as: $v_2=3\times10^{-6}$ m/s, $Q_{inj}=$ 30 m$^3$/d, $Q_{ext}$=-30
m$^3$/d, $\delta=2$ (negative skin), $\alpha_L$=0.1 m, $r_s$=0, 0.2, 0.4 and 0.6m. Fig. 16 shows the effect of the
skin thickness (negative skin) on BTCs during the pumping phase. At early stage, it can be
found that the concentration shows a decreasing trend with the increase of $r_s$, and the peak
values of BTCs also decrease with the increase of $r_s$. The explanation is similar to that for
Fig. 12 as a thicker negative skin means a thicker high conductance zone surrounding the test
well, which will facilitate the spreading of injected tracer mass further away from the test
well. This is further supported by the concentration distributions in a 2D horizontal plane 24
hr after the cease of injection (rest phase) with different negative skin thickness, as shown in
Fig.17. It is evident from Fig. 17 that a greater portion of tracer mass migrates away from the
test well after the cease of injection with a greater $r_s$. For instance, at early stage of extraction,
one can see that the BTC values in the case of $r_s$ =0.6 m are lower than those in the case of
$r_s$=0.2 m in Fig.16, and the tracer is transported further away from the test well in the case of
$r_s$ =0.6 m than that in the case of $r_s$=0.2 m in Fig.17.

**5. Conclusions**

In this study, a numerical model for a SWPP test with the presence of a regional
groundwater flow field, considering both the positive and negative skin effects was
investigated. There is a strong interaction between regional groundwater flow and well flow,



thus proper choices of the duration of each phase, and the injection and pumping rates should
be done in advance before the SWPP test to recollect the tracer as much as possible. Besides,
the numerical model of SWPP test can be used to obtain unknown parameters: i.e., regional
groundwater velocity, effective porosity, dispersivity, and biogeochemical reaction rates, by
fitting to the observed BTCs. The effects of both the hydraulic conductivities and thickness of
the skin zone on BTCs had also been considered. The following conclusions can be drawn:
1. Regional groundwater velocity has a significant effect on the shape of BTCs, a lower

regional groundwater velocity means that more tracer can be accumulated near the

symmetry plane around the well. The opposite is true for a case of a larger regional flow

velocity, resulting in a longer tailing of BTCs obtained during the extraction phase. In

addition, the pattern and location of the dividing streamline determine the quantity of tracer

mass extracted during the pumping phase.

2. We have proposed a skin index which is essentially the skin/formation hydraulic

conductivity ratio to quantify the skin impact. A lager skin index results in a lower

concentration for BTCs at early stage of pumping. On the contrary, a smaller skin index

means a higher concentration for BTCs at early stage of pumping. In addition, a smaller

skin index means that solute plume can accumulate more in the skin zone, otherwise, a

larger skin index results in a solute plume drifting further away from the skin zone after the

cease of the injection phase.

3. The impact of skin effect near a pumping well should not be neglected for SWPP test,

particularly when the regional groundwater flow is presented. The positive (or negative)

skin results in a faster (or lower) tracer transport process. A lager thickness of the positive



skin leads to a larger concentration of tracer near the symmetry plane around the well, but
the opposite is true for the case with a negative skin.
**Acknowledgements**

This research was partially supported by the National Natural Science Foundation of

China (Grant Numbers: 41772259, 41372253, 41521001), the Natural Science Foundation of
Hubei Province, China (2018CFA085,2018CFA028), the Fundamental Research Funds for
the Central Universities, China University of Geosciences (Wuhan).



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




**Figure Captions:**

**Figure Captions:**
Fig.1 The schematic diagram of the flow system.
Fig.2 Comparison between the numerical solutions of this study and the analytical solutions

of Huang et al. (2010).

Fig.3 BTCs for different values of $v_d$ at the well during the pumping phase.
Fig.4 The schematic diagram of the flow system for the pumping phase.
Fig.5 Concentration distributions in a 2D horizontal plane at $t_{pump}$=0 hr. a) $v_d = 1 \times 10^{-6}$ m/s,;

b) $v_d$ =1.5$\times 10^{-6}$m/s; c) $v_d$ =2$\times 10^{-6}$ m/s; d) $v_d = 3 \times 10^{-6}$ m/s.

Fig.6 BTCs for different values of $t_{res}$ at the well during the pumping phase.
Fig.7 Concentration distributions in a 2D horizontal plane at $t_{pump}$=0 hr. a) $t_{res} = 6$ hr; b) $t_{res} = $

126 hr; c) $t_{res} = 24$ hr; d) $t_{res} = 36$ hr.

Fig.8 BTCs at the well during the pumping phase with $\theta$=0.1, 0.2, 0.3, 0.4, 0.5.
Fig.9 BTCs at the well during the pumping phase with $\alpha_L$=0.01 m, 0.05 m, 0.1 m, 0.5 m.
Fig.10 BTCs for the case of a positive skin at the well during the pumping with $r_s$=0.6 m, and

$\delta$= 1, 0.5, 0.25 and 0.125.

Fig.11 2D horizontal plane distributions of concentration for a positive skin with $r_s$=0.6 m at

$t_{pump}$=0 hr. a) $\delta$= 1; b) $\delta$= 0.5; c) $\delta$= 0.25; d) $\delta$= 0.125.

Fig.12 BTCs for the case of a negative skin at the well during the pumping with $r_s$=0.6 m,

and $\delta$= 1, 1.5, 2 and 3.

Fig.13 2D horizontal plane distributions of concentration for a negative skin with $r_s$=0.8 m at

$t_{pump}$=0 hr. a) $\delta$= 1; b) $\delta$= 1.5; c) $\delta$= 2; d) $\delta$= 3.

Fig.14 BTCs for the case of a positive skin at the well during the pumping phase with $r_s$=0 m,

0.2 m, 0.4 m, 0.6 m.





Fig.15 Concentration distributions for a positive skin at 2D horizontal plane after 24 hr of

rest. a) $r_s$= 0 m; b) $r_s$= 0.2 m; c) $r_s$= 0.4 m; d) $r_s$= 0.6 m.

Fig.16 BTCs for the case of a negative skin at the well during the pumping with $r_s$=0 m,

0.2m, 0.4 m and 0.6 m.

Fig.17 Concentration distributions for a negative skin at 2D horizontal plane after 18 hr of

rest. a) $r_s$= 0 m; b) $r_s$= 0.2 m; c) $r_s$= 0.4 m; d) $r_s$= 0.6 m.






Table 1. The parameter values used in this study

| Parameter name | Symbols | Values |
| --- | --- | --- |
| Aquifer thickness (m) | $B$ | 10 |
| Radius of well screen (m) | $r_w$ | 0.1 |
| Density of groundwater($kg/m^3$) | $\rho$ | 1000 |
| Effective porosity of aquifer | $\theta$ | 0.3 |
| Hydraulic conductivity of aquifer (m/d) | $K$ | 8 |
| Constant heads of $S_1$(m) | $H_1$ | 15.22, 15.44, 15.69, 15.65, ,15.87, 16.08, 16.30 |
| Constant head of $S_2$ (m) | $H_2$ | 15.0 |
| Regional groundwater Darcy velocities (m/s) | $v_d$ | $5\times10^{-7}$, $1\times10^{-6}$, $1.5\times10^{-6}$, $2\times10^{-6}$, $2.5\times10^{-6}$, $3\times10^{-6}$ |
| Longitudinal dispersivities of aquifer (m) | $\alpha_L$ | 0.01, 0.05, 0.1, 0.5 |
| Hydraulic conductivity of skin zone (m/d) | $K_s$ | 1, 2, 4, 12, 16, 24 |
| Effective porosity of skin zone | $\theta_s$ | 0.21,0.24,0.27,0.33,0.36,0.39 |
| Injection or pumping rate ($m^3$/d) | $Q$ | 15, 30 |
| Mass flux per unit area ($kg/(m^2 \cdot s)$) | $N_0$ | 0.02765, 0.5529 |
| Injection time (hr) | $t_{inj}$ | 6 |
| Rest time (hr) | $t_{res}$ | 24 |

| Pumping time (hr) | $t_{pump}$ | 48 |
| --- | --- | --- |

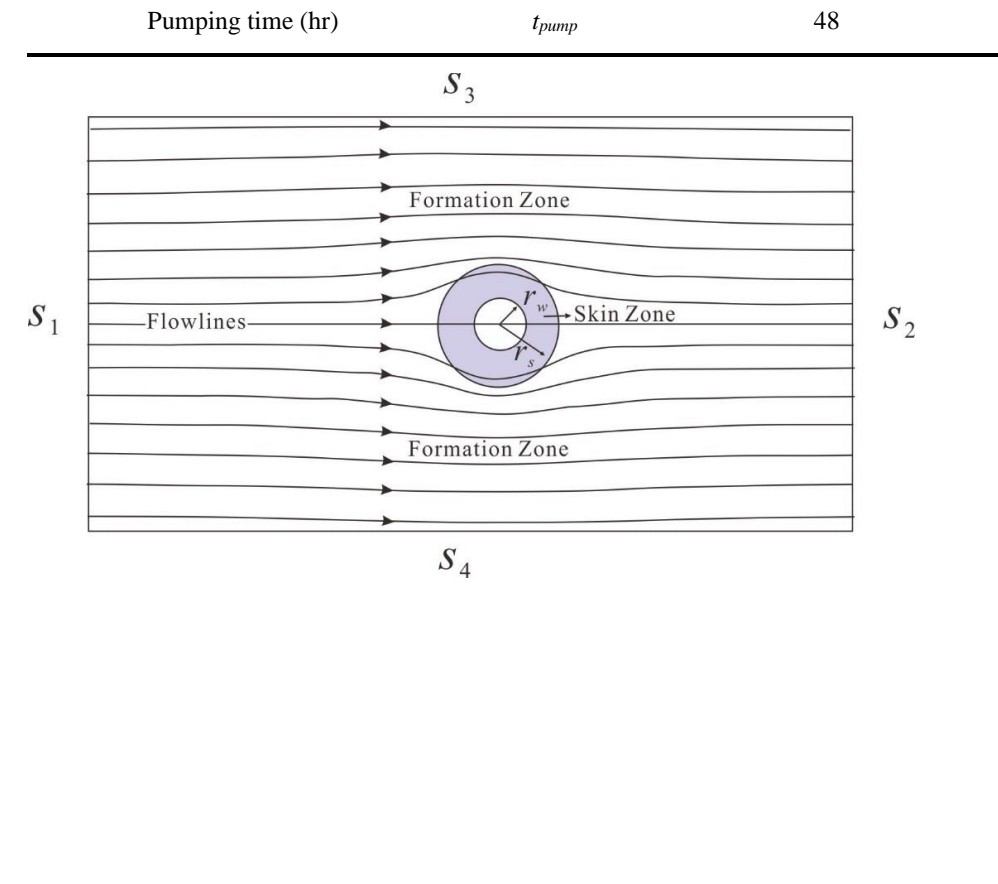






**Fig.1**





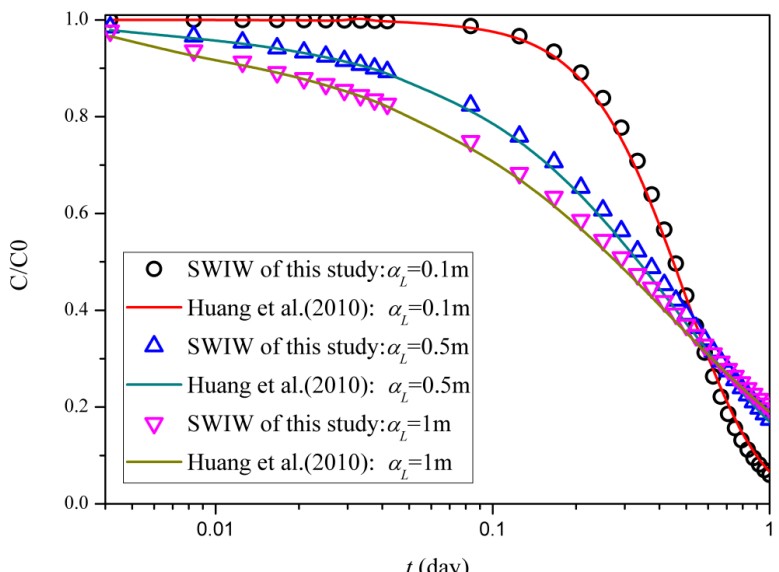






**Fig.2**





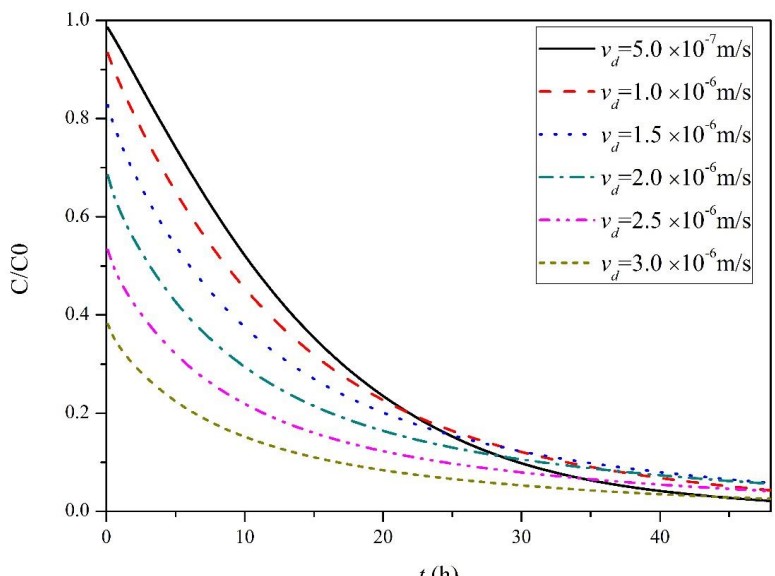





**Fig.3**



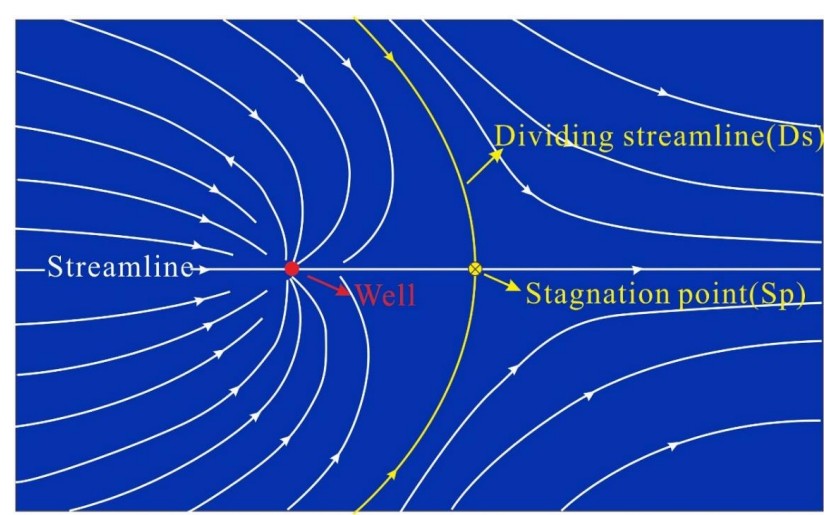







Fig.4



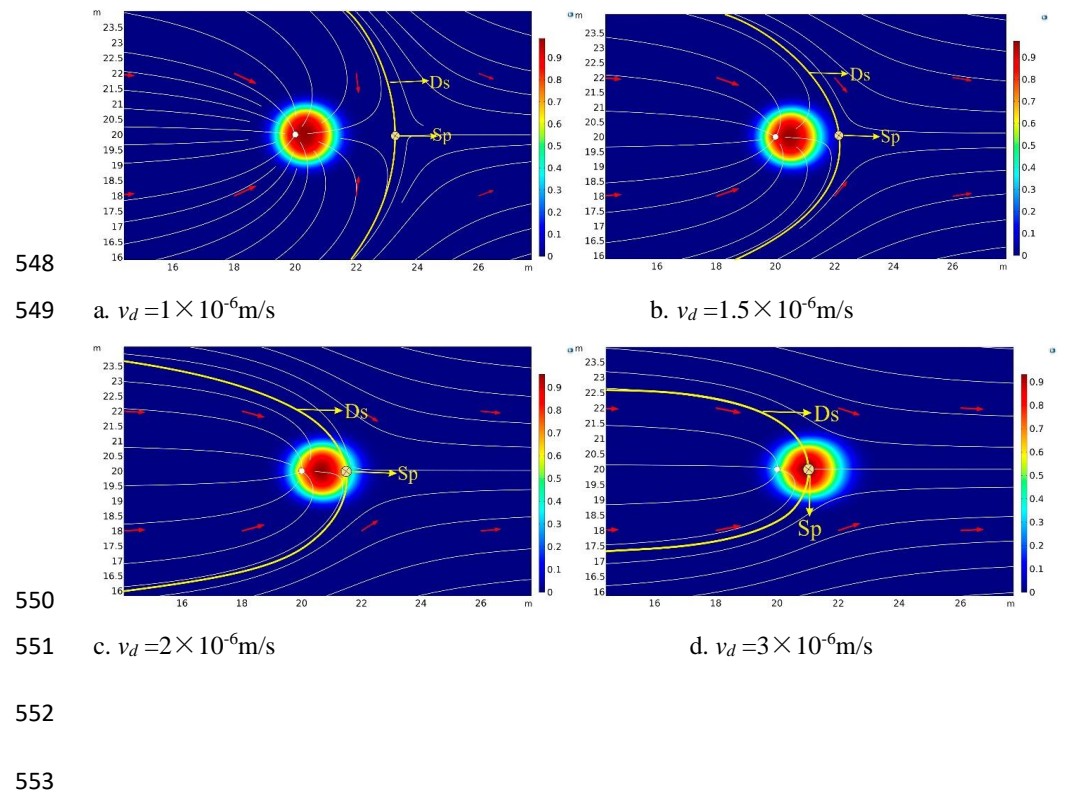


a. $v_d = 1 \times 10^{-6}$m/s                            b. $v_d = 1.5 \times 10^{-6}$m/s

c. $v_d = 2 \times 10^{-6}$m/s                            d. $v_d = 3 \times 10^{-6}$m/s



**Fig.5**




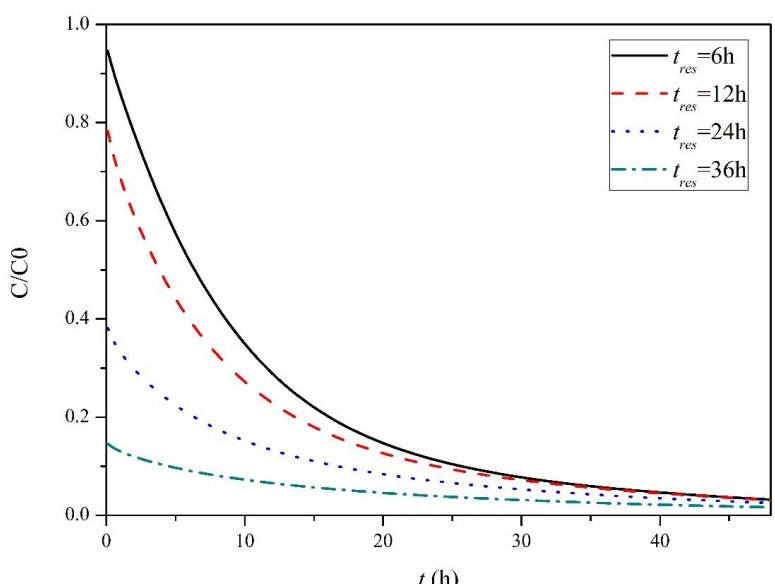




Fig.6




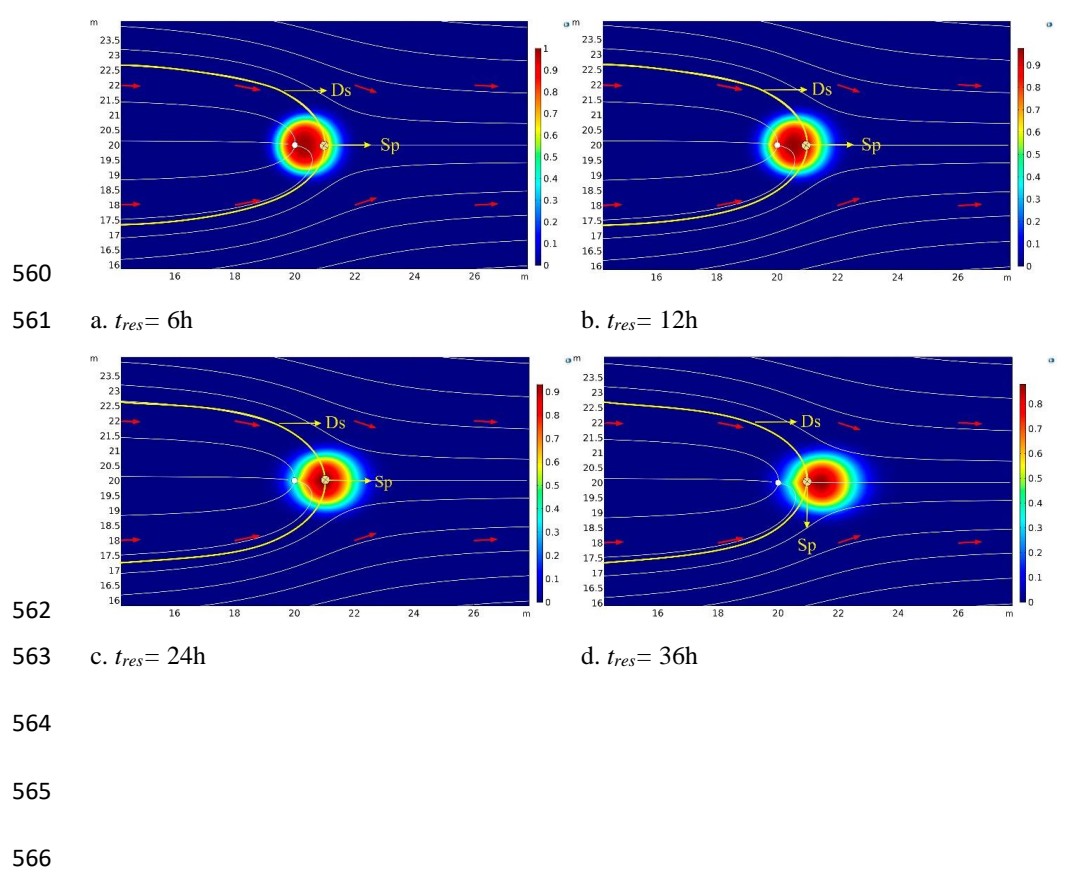


a. $t_{res}$= 6h                                    b. $t_{res}$= 12h

c. $t_{res}$= 24h                                   d. $t_{res}$= 36h




Fig.7



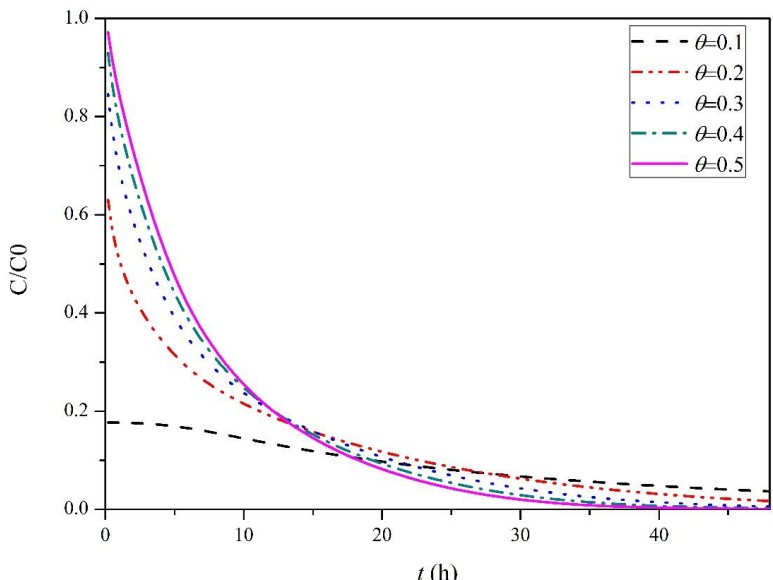




Fig.8






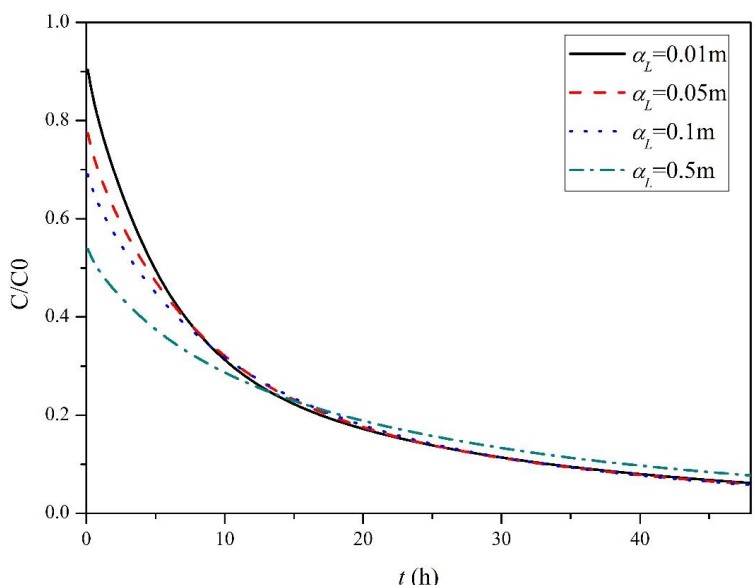




**Fig. 9**



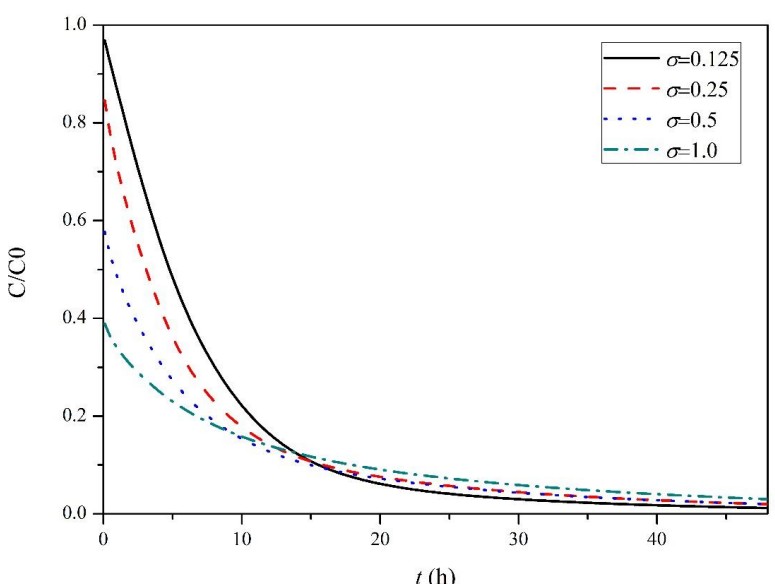






**Fig.10**



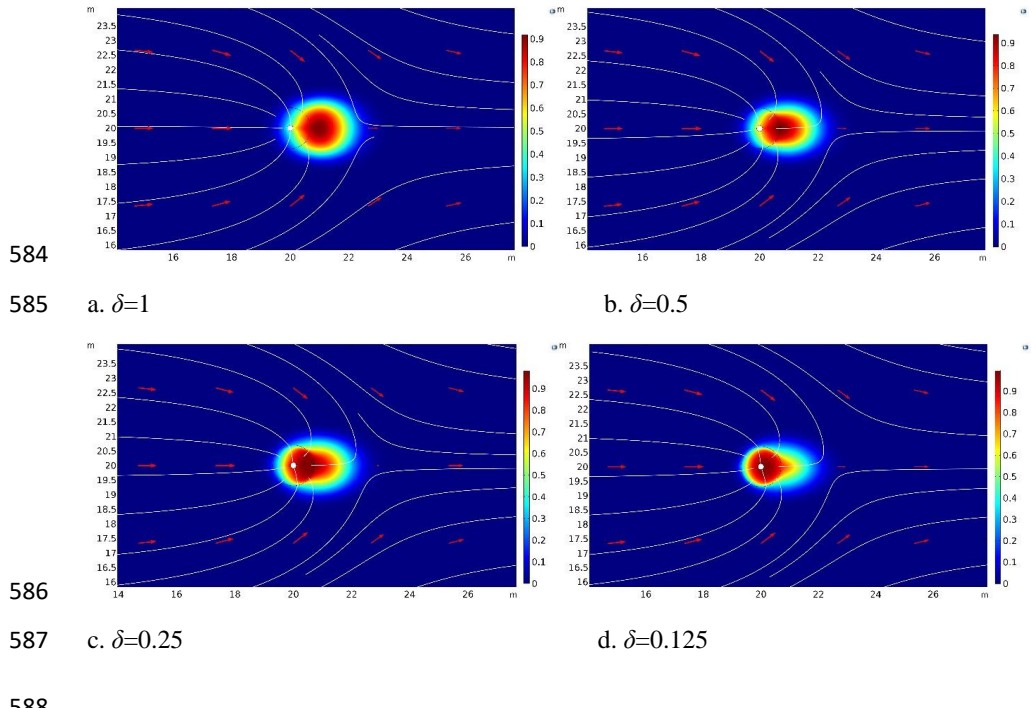


a. $\delta$=1                                        b. $\delta$=0.5

c. $\delta$=0.25                                    d. $\delta$=0.125

**Fig.11**





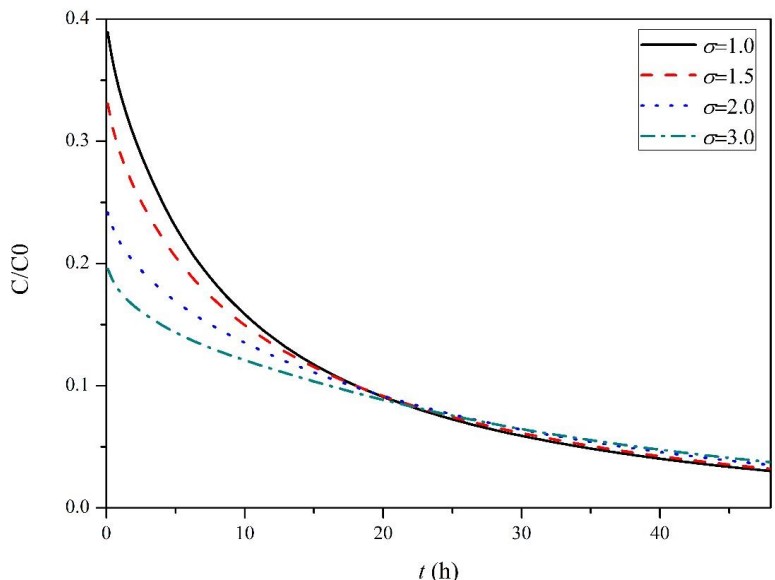





**Fig.12**




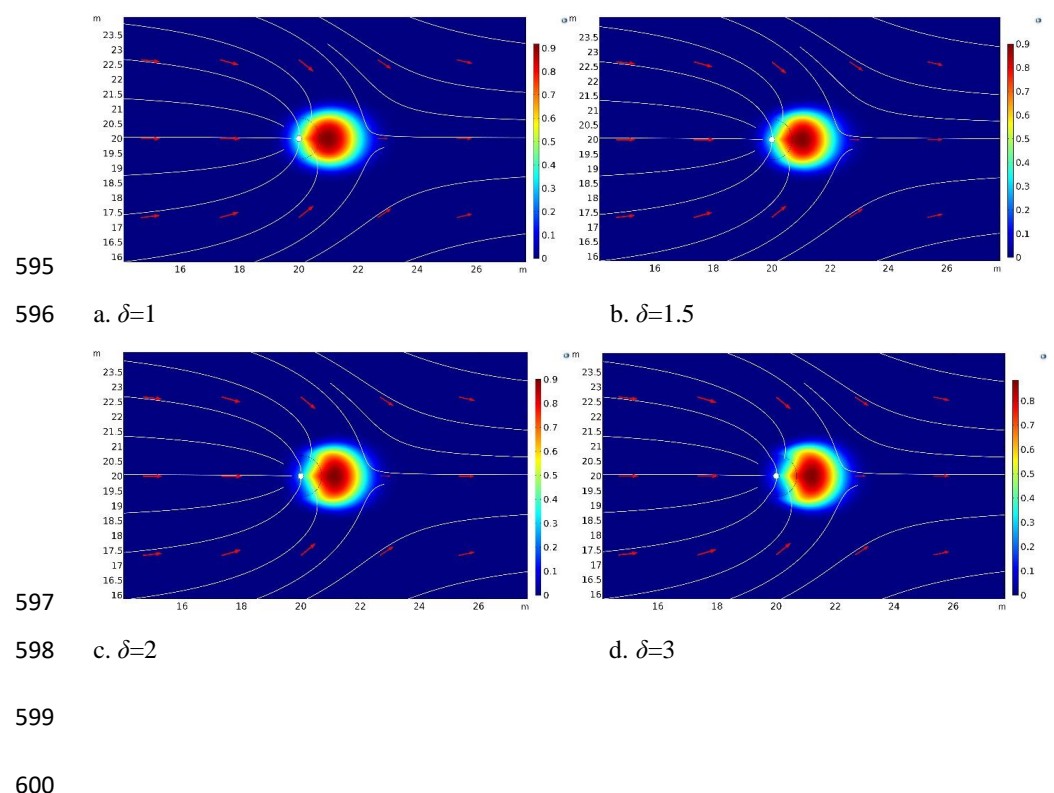


a. $\delta$=1                                               b. $\delta$=1.5

c. $\delta$=2                                               d. $\delta$=3


**Fig.13**





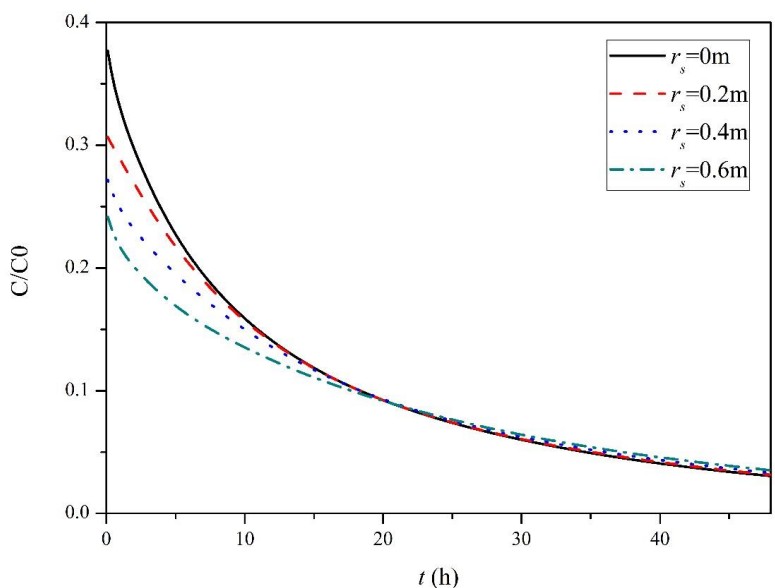





**Fig.14**





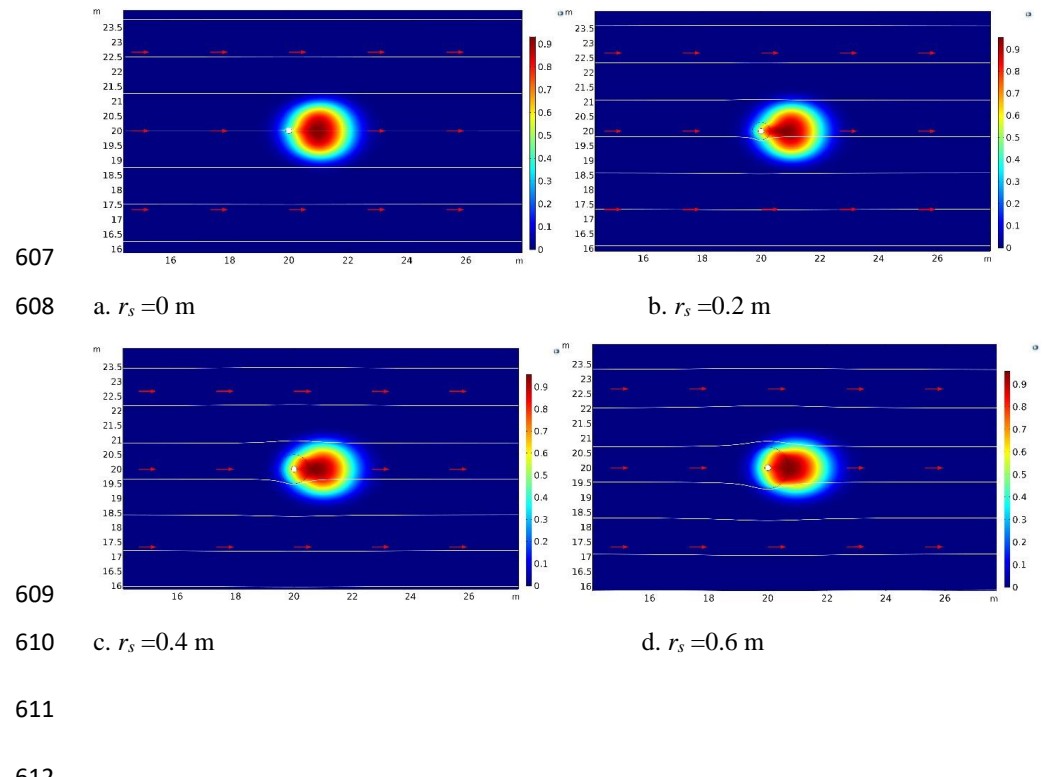


a. $r_s$ =0 m                                          b. $r_s$ =0.2 m

c. $r_s$ =0.4 m                                        d. $r_s$ =0.6 m



**Fig.15**





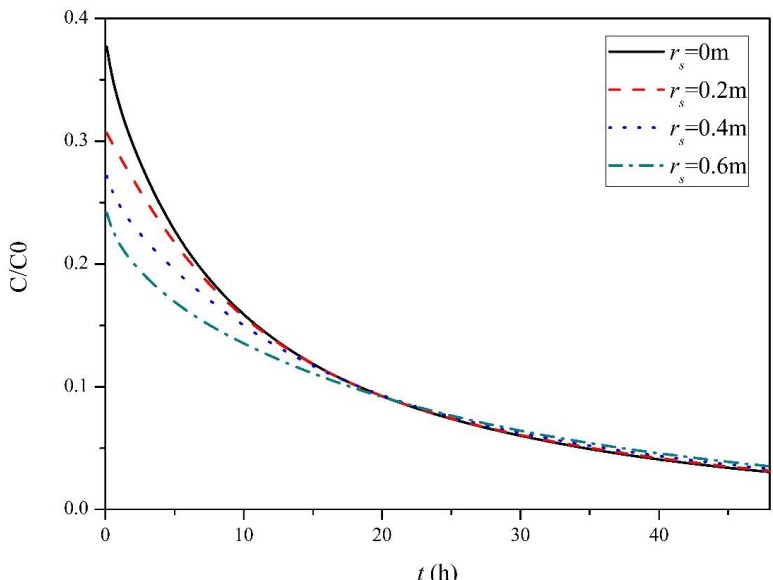





**Fig. 16**



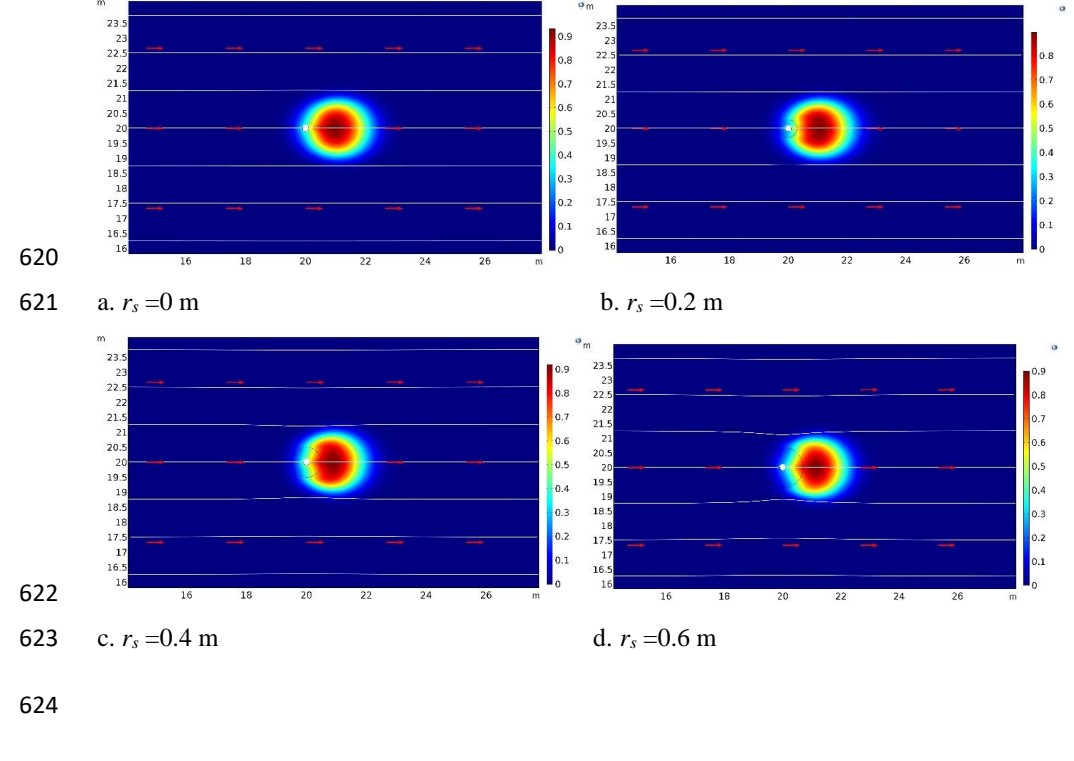


a. $r_s$ =0 m                                b. $r_s$ =0.2 m

c. $r_s$ =0.4 m                              d. $r_s$ =0.6 m


**Fig. 17**