# Peer review of "Manuscript under review for journal Hydrol. Earth Syst. Sci."

_Hydrology and Earth System Sciences, 2018_

## Referee Comment (RC1) · Anonymous Referee #1 · 26 Jul 2018

Comment on "Impact of skin effect on single-well push-1 pull tests with the presence of regional groundwater flow" by Xu Li, Zhang Wen, Hongbin Zhan and Qi Zhu

This research developed a numerical model for single-well push-pull test used to estimate aquifer parameters. The aquifer was conceptualized as a confined formation with a fully screened well involved. The work is significant, characterizing the flow and transport in a target aquifer. Some concerns are stressed below for the authors' reference. 1) Figure 4 looks NOT an appropriate flow pattern that satisfies the boundary conditions (6) [line 158, page 8], where the streamlines should be orthogonal to the upper and lower boundaries. The boundaries that are assumed to be no flux DO NOT behave this way. Please double check the model BC is set correctly. 2) During the "rest phase" ($t\_inj<t<t\_res$), there wouldn't be the well performance, but there still exits the

background groundwater flow which has the velocity v2>0, so the boundary condition (14) [line 193, page 10] was set inappropriately by ceasing the radial flux. It could be a good idea that setting no BC in the borehole at this phase. Some minor typos found: 1) Line 152, page 8, "r is the radial distance [L]" is repeatedly stated, previously its definition already given in line 147. 2) Line 158, notation "n" was not explained in context, it should be the norm vector of the boundary. 3) Line 206, page 11, the surface-integral over the borehole should be expressed more specifically, showing the integral variable (dr) under the integral sign.

---

## Referee Comment (RC2) · Anonymous Referee #2 · 18 Sep 2018

Review for HESS – hess-2018-279

Title: Impact of skin effect on single-well push-pull tests with the presence of regional groundwater flow

by Xu Li et al.

General comments:

Groundwater tracer experiments are an important tool for the in-situ assessment of aquifer physical, chemical, and biological properties. Among other techniques, single-well push-pull tests (PPTs) have received considerable attention over the past decades for in-situ assessment of aquifer characteristics. Early PPT papers dealing with the determination of regional groundwater flow velocity and porosity (Leap and Kaplan,

1988; Hall et al., 1991) mentioned a "velocity shadow" downgradient of the pumping well, which may adversely affect the estimation of these parameters. However, this issue has not been quantitatively addressed in the literature to date.

In their current manuscript, the authors attempt to fill this gap by producing numerical simulations of PPTs in the presence of a skin effect under regional groundwater flow conditions. As such, I see merit in this manuscript, as it would provide scientists and practitioners with important information on the accuracy of parameters obtained from PPTs conducted under these particular conditions. On the other hand, I see several important shortcomings in this manuscript, which need to be addressed before it may become suitable for publication. My main concerns are listed here, detailed issues are in the specific comments section below.

1. The manuscript currently lacks conciseness in writing and a careful review of the pertinent (including recent) literature (see specific comments 8 and 9). As suggested by the title, the focus should be on the effect of skin effects on PPTs, because this issue has not been addressed quantitatively before. But as is, the results of the COM-SOL simulations are presented in an excessively large number of figures. The authors should carefully consider which figures are essential to providing new insights into the skin effect during PPTs (i.e., the main objective of their paper), and consider combining these figures whenever possible. Unrelated figures (e.g., effect of aquifer effective porosity, dispersivity, etc. on PPT breakthrough curves) should be deleted or moved to a supplementary information section.

2. The simulation results are presented in "qualitative" fashion only, i.e., the reader can only visually compare the breakthrough curves and 2-d spatial concentration distributions between different simulations to judge the effect and relevance of the skin effect. To allow for a more quantitative comparison between simulations, the authors could, e.g., compute relative tracer mass recovered by the end of each PPT, or provide a moment analysis for mass distribution in the 2-d plots. In addition, the presented results are conditional with respect to the simulated scenarios. For readers to apply these re-

sults in their own work, a more general (dimensionless) analysis of PPT breakthrough curves would be preferable.

3. An important deficiency of the current manuscript is that the authors never go beyond presenting PPT breakthrough curves and 2-d spatial concentration patterns as affected by skin effects. The central question, how the skin effect affects the estimation of aquifer properties such as regional groundwater flow velocity and porosity estimated from PPTs (which is why PPTs are conducted in the first place), remains unanswered. Without such information, the reader cannot judge the importance of this phenomenon on the results presented in this manuscript, and the relevance of skin effects during PPTs in general. Quantitative information on this issue could be provided, e.g., by applying the model of Hall et al. (1991) to simulation PPT breakthrough curves in an attempt to recover values for regional groundwater flow velocity and porosity, and to compare the latter with respective simulation input values.

4. The current writing style is poor and improvements need to be made both with regard to sentence/paragraph structure as well as grammar. The manuscript should be edited by a native English speaker.

Specific comments:

1. l. 18-33: Abstract: I am afraid that the abstract is not very informative to a general audience, as it is full of unexplained, specific terminology that only an insider to the subject matter may understand. Examples are "dividing streamline", "skin", "positive skin", "negative skin".

2. l. 22: The sentence "In this study, a new numerical model . . .was established" is misleading. The authors used/adapted the commercially available COMSOL code/model to simulate PPTs in a confined aquifer under regional groundwater flow in the presence of skin effects. They did not develop a new numerical (finite-element) model.

3. l. 39: Here the authors describe PPTs as two-stage (injection/extraction) experiments. Several lines below (l. 43) they revisit this subject and state that a PPT may contain four phases (tracer injection, chaser injection, rest and pumping). Why not combine the two and say from the beginning that PPTs may consist of up to four phases? This would avoid confusion and redundancy.

4. l. 44: The term "rest phase" is an unfortunate terminology in the context of this manuscript. Although I am aware that this term is used in some of the PPT literature, the PPT literature dealing with determination of groundwater flow velocity and porosity prefers the term "drift phase". The latter term much better reflects the conditions encountered under regional groundwater flow conditions. In addition, whereas the authors mention that "the rest phase is for tracer to diffuse and/or react with the aquifer (if a reactive tracer is employed)", they fail to mention here that such a drift phase is crucial for the determination of groundwater flow velocity and porosity (Leap and Kaplan, 1988; Hall et al., 1991).

5. l. 73-74: In light of previous findings (e.g., Vandenbohede et al., 2008), I believe that the statement regarding determination of regional groundwater flow velocity is not really supported in recent literature.

6. l. 75: Why a three-well minimum? A gradient may be obtained from two wells given that they are aligned in groundwater flow direction. A better explanation should be provided.

7. l. 84: Here the authors return to explaining PPTs (see comment 3), and now mention three phases. This is confusing and redundant. Why not combine with previous sections (l. 39/44)?

8. l. 90: "that if the solute transport drifted over the location of dividing streamline toward downstream". First, it is unclear what is meant by "dividing streamline". Whenever new terminology is introduced, it should be explained to readers at the first instance it is used. Second, more importantly, and to the best of my knowledge, this is not what Leap and Kaplan (1988) have reported! They do not mention a dividing streamline

(beyond which no solute can be recovered), rather they mention a "velocity shadow" downgradient of the well, in which "advective velocity may be slightly less than at a greater distance downstream…". In other words, this is the first mentioning of a "skin effect" during PPTs. Hall et al. (1991) later pick up on this issue of a "velocity shadow". Conversely, Monkmeyer and Netzer (1993) in their comment on Leap and Kaplan's 1998 paper, appear to be the first to consider a dividing streamline and a stagnation point during a PPT pumping phase (see Fig. 1b in Monkmeyer and Netzer, 1993).

9. l. 96: In their review of literature dealing with the determination of regional groundwater flow velocity and/or porosity during PPTs, the authors may want to include recent publications, e.g., by Paradis et al. (2018), Hansen et al. (2016, 2017), Johnsen and Whitson (2009).

10. l. 127: "…so that the wellbore effect is not a concern." This statement is unclear. The authors should explain "wellbore effect". Do they mean wellbore storage? Again, when new terminology is introduced, it needs explanation at the first instance of use.

11. l. 131: The authors mention the coordinate system used and refer to Fig. 1. But why is the coordinate system not depicted in Fig. 1?

12. l. 134 and 163: Mathematical model of flow and transport: It is not clear to me why the authors present a mathematical model here, and what is new about this model. The flow model does not include equations that would take into account the skin effect in an analytical fashion, not is the model later used to quantitatively assess the COMSOL numerical output. The same holds true for the transport model. Boundary and initial conditions are of course needed to explain the COMSOL simulations performed by the authors, but they could be presented in chapter 3.

13. l. 158: Is parameter "n" in eq. 6 explained in the text? I couldn't find it.

14. l. 180: "..the inner boundary condition inside the well..". It is unclear to which boundary the authors refer to. The well casing?

15. l. 189: "During the rest phase, the solute flux from the borehole into the aquifer is zero,...". I don't agree with this statement. Given that the borehole has a finite dimension in the authors' simulations, there should be solute mass contained in the borehole at the end of the injection phase, and thus at the beginning of the rest phase. This solute should get flushed out of the borehole by regional groundwater flow.

16. Table 1: I couldn't find the skin radius rs in Table 1. Is there a reason not to list it?

17. l. 228: "..progressively refined near the well." It remains unclear how fine the mesh size actually was near the well. Readers wanting to repeat the simulations will need to know.

18. l. 239: Results: (1) The results of the COMSOL simulations are presented in an excessively large number of figures. The authors should carefully consider which figures are essential to providing new insights into the skin effect during PPTs (the main objective of their paper), and consider combining figures whenever possible. For example, Figs. 14 and 16 show PPT breakthrough curves affected by positive and negative skin effects. These two figures could easily be combined into a single figure. Other figures not immediately related to the main objective should be deleted or may be moved to a supplementary document. (2) The results are presented in qualitative fashion only, i.e., the reader can only visually compare the breakthrough curves between different simulations to judge tracer mass recovery. To improve this comparison the authors could, e.g., compute relative tracer mass recovered by the end of each PPT. This would allow for a more quantitative comparison.

19. l. 250: ".. one can see that there is a stagnation point (Sp) located at the dividing streamline (Ds) as shown in Fig.4." This statement and figure are correct, but not new (see Monkmeyer and Netzer, 1993). Also, the term stagnation point is introduced without an explanation. What is the relevance of the stagnation point?

20. l. 270: the effect of resting time: The results of this section are a logical consequence of results from the previous section, where the effect of regional groundwater

flow velocity are shown. Therefore, I would suggest to shorten this section and combine it with the previous.

21. l. 348: "Fig. 14 shows the effects of the skin thickness (positive skin) on BTCs during the pumping phase. One can see that the concentration gets higher at early stage with the increase of rs." This is not what is shown in Fig. 14, but rather the opposite (shown is highest early-time conc. for rs = 0). In fact, data plotted in Figs. 14 and 16 look identical. I suspect that the wrong set of data was plotted in Fig. 14. Furthermore, this is another example of two figures which could easily be combined into a single figure.

22. l. 381: "Besides, the numerical model of SWPP test can be used to obtain unknown parameters: i.e., regional groundwater velocity, effective porosity, dispersivity, and biogeochemical reaction rates, by fitting to the observed BTCs." I find this conclusion unwarranted based on the merely qualitative results provided. First, inverse modeling is not a new element, this has been done before to assess parameters from PPTs (e.g., Gelhar and Collins, 1971, Schroth et al., 2001, Vandenbohede et al., 2008). But more importantly, the authors have not provided any data or sensitivity analysis for this approach in their manuscript. It remains therefore unknown (and questionable) if such an inverse modeling approach will yield unique parameters sets with sufficient accuracy.

---

## Referee Comment (RC3) · Anonymous Referee #3 · 23 Sep 2018

General comments

The objective of this paper is to study the impacts of skin effects on Single Well Push-Pull (SWPP) tests for estimating groundwater flow velocity. To this aim, the authors build a numerical model of a SWPP test with skin effects using COMSOL. The model is developed in steady-state condition for a 2D homogeneous aquifer. They validate the numerical model against an analytical solution in the case of a SWPP test with no skin effect and no groundwater flow. The authors show the effects on breakthrough curves (BTCs) of the following quantities: groundwater flow velocity, duration of the rest phase, porosity, dispersivity, hydraulic conductivity of the skin zone, thickness of the skin zone. They conclude the following points: groundwater flow velocity should be considered in order to properly design a SWPP test so that as much tracer as possible

is recovered; skin effects impact BTCs significantly and should be considered during inverse modelling of SWPP tests in order to estimate the correct flow and/or transport parameters.

The manuscript is generally clearly written. The subject is interesting and useful to the readers of HESS. Indeed, it points out that skin effects should not be neglected when interpreting the results of a SWPP test for parameter estimation, i.e., skin effects must be considered in the mathematical model used to fit BTCs. However, I think that authors should clarify some points, as explained in the following specific and technical comments.

Specific comments:

1. According to lines 114-116, the objective of this work is to study the impacts of skin effects on SWPP tests for estimating groundwater flow velocity. However, the article only shows that the skin effect has a significant influence on the shape of the BTC and on the 2D distribution of concentration. It does not quantify the impacts of skin effects on SWPP tests "for estimating groundwater velocity". Knowing the impact of skin effects on the estimation of groundwater flow velocity (and of transport parameters) is of primary importance for practical applications. For example, in section 4.1 do the authors manage to estimate correctly the velocity in all cases? Ideally the authors should estimate groundwater flow velocity in the different skin configurations and compare it to the real one. At least, they should reformulate the objective of the paper and give an indication of the expected error on the identified groundwater flow velocity.

2. I am not sure to understand the reason why the authors show the impact of groundwater flow velocity on BTCs (section 4.1). Groundwater flow velocity affects the BTC and that is why SWPP tests can be used to estimate this parameter. It is interesting to better understand how groundwater flow velocity influences the SWPP test. However, it is not clear how this analysis is related to the objective of the paper (assessing the impacts of skin effects on SWPP tests). In the conclusions, the authors points out that

groundwater flow velocity should be considered in order to design the experiment so that as much tracer as possible is recovered. This could be the reason that justifies section 4.1. Nevertheless, it is not explained how it is related to the objective of the paper.

3. The same remark of point 2. can be done for the analysis of the impact of the duration of the rest phase, of porosity and dispersivity (sections 4.2, 4.3, 4.4).

4. Too many figures are presented: I suggest the authors to choose only the figures which are relevant to the objective of the article. In my opinion, figures 6, 7, 8, 9 are not necessary. Moreover, figures showing results with positive skin effects could be combined with figures showing results with negative skin effects.

Technical corrections:

l.23 "the finite-element COMSOL Multiphysics": add "software".

l.27 Dividing streamline: this sentence becomes clear only after having read the article. The authors should explain what they mean by "dividing streamline".

l.29-30 I think the sentence is not very clear. It could be reformulated as: "a smaller ratio between the hydraulic conductivity of the positive skin and that of the aquifer formation". Moreover, there is no need here to write the mathematical symbol \delta.

Figure 1 The term "formation zone" is not very clear to me. Maybe it can be changed with "aquifer" or "aquifer formation". The caption should precise what are S1, S2, S3, S4. The coordinate axes are missing.

l.127 What is the "wellbore effect"? If this is important for the understanding of the paper it should be explained, otherwise it can be removed.

l.152 r was already defined at l.147.

l.159 I think it should be specified that H is head.

l.200 and l.70 Wang et al (2017) is missing in the references section

l.270 In the section title, symbol tres should be changed with "the duration of the rest phase"?

Figures 10 and 12: The symbol should be $\delta$ and not $\sigma$.

Figures 11, 13 Why some of the flow lines are interrupted?

l.331 "In contract to": I would rather say "In agreement with"

Figure 14 is probably wrong: concentration decreases with rs, differently from what is said at line 350.

Figures 11, 13, 15 and 17: The skin zone is not very evident in the figures. Maybe it could be highlighted with a thicker line.

Table 1: the skin thickness default value is missing.

―――――――――――――――――――――――

---

## Author Comment (AC1) · 24 Oct 2018

Thank the reviewer very much for his/her careful check on the manusript. The point to point response can be found in the following. Please note that the referred page number and line number are referred to the marked version in the supplement files. The supplement files include: a marked version of the revised manuscript, a clean version of the revised manuscript, a supplementary material of the revised manuscript, and the response letter to all the referees' comments.

1) Figure 4 looks NOT an appropriate flow pattern that satisfies the boundary conditions (6) [line 158, page 8], where the streamlines should be orthogonal to the upper and lower boundaries. The boundaries that are assumed to be no flux DO NOT behave

this way. Please double check the model BC is set correctly Reply: We are sorry for not making this point clearly in the previous manuscript. Actually, the streamlines are orthogonal to the upper and lower boundaries in this model, Figure 4 only shows a flow pattern for a small area nearby the well, not for the entire domain, thus the streamlines there appear not orthogonal to the upper and lower boundaries. We have added "nearby the well" in the figure caption and also clarify this point in the revised manuscript. See p. 18 lines 356-358. 2) During the "rest phase" (t_inj<t<t_res), there wouldn't be the well performance, but there still exits the background groundwater flow which has the velocity v2>0, so the boundary condition (14) [line 193, page 10] was set inappropriately by ceasing the radial flux. It could be a good idea that setting no BC in the borehole at this phase. Reply: We agree with the reviewer on this point. We have revised it accordingly as the reviewer suggested. See p. 13, lines 251-257. Some minor typos found: 1) Line 152, page 8, "r is the radial distance [L]" is repeatedly stated, previously its definition already given in line 147. Reply: Implemented. See p. 10, line 192. 2) Line 158, notation "n" was not explained in context, it should be the norm vector of the boundary. Reply: The notation "n" has been explained in manuscript. See p. 15, lines 281-282. 3) Line 206, page 11, the surface-integral over the borehole should be expressed more specifically, showing the integral variable (dr) under the integral sign. Reply: Actually, this intergral variable should be d$\theta$ instead of dr. The range of $\theta$ is [0, $2\pi$], meaning that this equation is integrated over the perimeter of the borehole. After a serious consideration, we think it is more appropriate to keep it as it was. See p. 17, line 323

Please also note the supplement to this comment:
https://www.hydrol-earth-syst-sci-discuss.net/hess-2018-279/hess-2018-279-AC1-supplement.zip

---

## Author Comment (AC2) · 24 Oct 2018

Thank the reviewer very much for his/her careful check on the manusript. The point to point response can be found in the following. Please note that the referred page number and line number are referred to the marked version in the supplement files. The supplement files include: a marked version of the revised manuscript, a clean version of the revised manuscript, a supplementary material of the revised manuscript, and the response letter to all the referees' comments.

1. The manuscript currently lacks conciseness in writing and a careful review of the pertinent (including recent) literature (see specific comments 8 and 9). As suggested by the title, the focus should be on the effect of skin effects on PPTs, because this

issue has not been addressed quantitatively before. But as is, the results of the COMSOL simulations are presented in an excessively large number of figures. The authors should carefully consider which figures are essential to providing new insights into the skin effect during PPTs (i.e., the main objective of their paper), and consider combining these figures whenever possible. Unrelated figures (e.g., effect of aquifer effective porosity, dispersivity, etc. on PPT breakthrough curves) should be deleted or moved to a supplementary information section.

Reply: The conciseness in writing and a careful review of the pertinent literature have been strengthened in this revised manuscript (see replies for specific comments 8 and 9). In addition, the unrelated figures (e.g., effect of aquifer effective porosity, dispersivity, etc. on PPTs breakthrough curves) have been moved into a supplementary material as references. In this revised manuscript, we have analyzed the effect of skin effects on parameter estimations thoroughly as the reviewer suggested (see section 4.5).

2. The simulation results are presented in "qualitative" fashion only, i.e., the reader can only visually compare the breakthrough curves and 2-d spatial concentration distributions between different simulations to judge the effect and relevance of the skin effect. To allow for a more quantitative comparison between simulations, the authors could, e.g., compute relative tracer mass recovered by the end of each PPT, or provide a moment analysis for mass distribution in the 2-d plots. In addition, the presented results are conditional with respect to the simulated scenarios. For readers to apply these results in their own work, a more general (dimensionless) analysis of PPT breakthrough curves would be preferable.

Reply: In this revised manuscript, to allow for a more quantitative comparison between simulations, we have computed the relative tracer mass recovered at the end of each PPT and provided a moment analysis for mass distribution versus distance with different skin properties (e.g., skin hydraulic conductivity and skin thickness), as the reviewer suggested. See sections 4.2, 4.3 and 4.4. For the dimensionless analysis, it is usually preferable for analytical modeling as it can reduce the number of variables, thus help

gain better insights in system analysis. However, for the numerical modeling like this paper, one needs to set all the (dimensional) values for the parameters that are representatives of realistic situations. After a careful consideration of this comment, we still think it is better to use dimensional variables for the numerical analysis.

3. An important deficiency of the current manuscript is that the authors never go beyond presenting PPT breakthrough curves and 2-d spatial concentration patterns as affected by skin effects. The central question, how the skin effect affects the estimation of aquifer properties such as regional groundwater flow velocity and porosity estimated from PPTs (which is why PPTs are conducted in the first place), remains unanswered. Without such information, the reader cannot judge the importance of this phenomenon on the results presented in this manuscript, and the relevance of skin effects during PPTs in general. Quantitative information on this issue could be provided, e.g., by applying the model of Hall et al. (1991) to simulation PPT breakthrough curves in an attempt to recover values for regional groundwater flow velocity and porosity, and to compare the latter with respective simulation input values.

Reply: In this revised manuscript, we have added a section about parameter estimations, and analyzed how different is the estimated parameters (e.g. dispersivity, porosity and regional groundwater velocity) based on the model without skin from their "actual" values based on the flow model with skin. The results indicate that the parameters estimated by the non-skin model are quite different from the real values, resulting in larger errors in estimating those parameters. After a careful check of the solution of Hall et al. (1991), we found that it was not rigorously derived with some important details either missing or unexplained. For example, the transport of the tracer during the push phase was negligible (Charles et al., 2018). Thus, the applicability of Hall et al. (1991) is questionable and requires further scrutiny. Based on above considerations, we decide not to use the model of Hall et al. (1991) to simulate PPT breakthrough curves. See section 4.5 for more details in the revised manuscript.

4. The current writing style is poor and improvements need to be made both with

regard to sentence/paragraph structure as well as grammar. The manuscript should be edited by a native English speaker.

Reply: The writing style has been improved, and English of the paper has been polished.

Specific comments: 1. l. 18-33: Abstract: I am afraid that the abstract is not very informative to a general audience, as it is full of unexplained, specific terminology that only an insider to the subject matter may understand. Examples are "dividing streamline", "skin", "positive skin", "negative skin".

Reply: The specific terminology (e.g. skin, positive skin, negative skin) have been explained in the abstract. In addition, terminology "dividing streamline" has been revised. See p. 2, lines 20-35.

2. l. 22: The sentence "In this study, a new numerical model . . . was established" is misleading. The authors used/adapted the commercially available COMSOL code/model to simulate PPTs in a confined aquifer under regional groundwater flow in the presence of skin effects. They did not develop a new numerical (finite-element) model.

Reply: The word "new" has been deleted. See p. 2, line 26.

3. l. 39: Here the authors describe PPTs as two-stage (injection/extraction) experiments. Several lines below (l. 43) they revisit this subject and state that a PPT may contain four phases (tracer injection, chaser injection, rest and pumping). Why not combine the two and say from the beginning that PPTs may consist of up to four phases? This would avoid confusion and redundancy.

Reply: To avoid confusion and redundancy, we have stated at the beginning that the PPTs consist of two phases. See p. 4, lines 47-55.

4. l. 44: The term "rest phase" is an unfortunate terminology in the context of this manuscript. Although I am aware that this term is used in some of the PPT literature, the PPT literature dealing with determination of groundwater flow velocity and porosity prefers the term "drift phase". The latter term much better reflects the conditions encountered under regional groundwater flow conditions. In addition, whereas the authors mention that "the rest phase is for tracer to diffuse and/or react with the aquifer (if a reactive tracer is employed)", they fail to mention here that such a drift phase is crucial for the determination of groundwater flow velocity and porosity (Leap and Kaplan, 1988; Hall et al., 1991).

Reply: The term "rest phase" has been replaced with "drift phase" in the manuscript. See p.7, lines 117-118.

5. l. 73-74: In light of previous findings (e.g., Vandenbohede et al., 2008), I believe that the statement regarding determination of regional groundwater flow velocity is not really supported in recent literature.

Reply: We have revised it, and its application for determining the regional groundwater velocity has rarely been discussed in previous studies, thus we have provided some in-depth discussion about this matter. See p. 5-6, lines 82-106.

6. l. 75: Why a three-well minimum? A gradient may be obtained from two wells given that they are aligned in groundwater flow direction. A better explanation should be provided.

Reply: The groundwater flow velocity may be measured directly using a two-well tracer test conducted under nature gradient condition, but this requires a monitoring well that is located directly down-gradient at a convenient distance from the test well, which is unlikely in most field applications (as one may not be aware of the hydraulic gradient and groundwater flow direction before the installation of monitoring wells). In fact, in most cases, the hydraulic gradient is determined using a three-well triangle in a homogeneous aquifer, and the groundwater flow velocity (including its magnitude and direction) may be obtained if the hydraulic conductivity and effective porosity are also known. If the hydraulic conductivity and effective porosity are unavailable, one may rely on the BTCs obtained from such a three-well system in a natural gradient tracer test as

an alternative to determine the regional flow velocity and longitudinal and transverse dispersivities as well. This can be done using the following procedures. First, the direction of hydraulic gradient can be determined based on the hydraulic head measurements in three monitoring wells, and the groundwater flow direction is directly opposite of the hydraulic gradient direction in a horizontally isotropic media (which is usually true for most field applications). Second, after determining the direction of groundwater flow, now one has three more parameters to determine: the magnitude of the groundwater flow velocity and longitudinal and transverse dispersivities. Such three unknown parameters can be obtained using the concentrations measured in above three observation wells. See p. 5-6, lines 84-106.

7. l. 84: Here the authors return to explaining PPTs (see comment 3), and now mention three phases. This is confusing and redundant. Why not combine with previous sections (l. 39/44)?

Reply: We have combined this part with previous sections. To estimate aquifer parameters such as porosity, dispersivity, biogeochemical reaction rate, etc., a two-phase PPT (tracer injection and pumping) will suffice to meet the need. However, if we want to estimate regional groundwater flow velocity, we need to add a drift phase in addition to the injection and pumping phases. See p. 4, lines 44-55 and p. 7, lines 116-119.

8. l. 90: "that if the solute transport drifted over the location of dividing streamline toward downstream". First, it is unclear what is meant by "dividing streamline". Whenever new terminology is introduced, it should be explained to readers at the first instance it is used. Second, more importantly, and to the best of my knowledge, this is not what Leap and Kaplan (1988) have reported! They do not mention a dividing streamline (beyond which no solute can be recovered), rather they mention a "velocity shadow" downgradient of the well, in which "advective velocity may be slightly less than at a greater distance downstream...". In other words, this is the first mentioning of a "skin effect" during PPTs. Hall et al. (1991) later pick up on this issue of a "velocity shadow". Conversely, Monkmeyer and Netzer (1993) in their comment on Leap and Kaplan's

1998 paper, appear to be the first to consider a dividing streamline and a stagnation point during a PPT pumping phase (see Fig. 1b in Monkmeyer and Netzer, 1993).

Reply: Implemented. Firstly, the terminology "dividing streamline" has been explained. Secondly, "velocity shadow" and "stagnation point" in previous literatures have been described again. See p. 7-8, lines 121-136.

9. l. 96: In their review of literature dealing with the determination of regional groundwater flow velocity and/or porosity during PPTs, the authors may want to include recent publications, e.g., by Paradis et al. (2018), Hansen et al. (2016, 2017), Johnsen and Whitson (2009).

Reply: Those recent literatures have been added into introduction. See lines 47, 74 and 130-132.

10. l. 127: ". .: so that the wellbore effect is not a concern." This statement is unclear. The authors should explain "wellbore effect". Do they mean wellbore storage? Again, when new terminology is introduced, it needs explanation at the first instance of use.

Reply: The "wellbore effect" has been replaced with "wellbore storage". See p. 9, line 168.

11. l. 131: The authors mention the coordinate system used and refer to Fig. 1. But why is the coordinate system not depicted in Fig. 1?

Reply: The coordinate system has been depicted in Fig. 1. See p. 43, Fig. 1.

12. l. 134 and 163: Mathematical model of flow and transport: It is not clear to me why the authors present a mathematical model here, and what is new about this model. The flow model does not include equations that would take into account the skin effect in an analytical fashion, not is the model later used to quantitatively assess the COMSOL numerical output. The same holds true for the transport model. Boundary and initial conditions are of course needed to explain the COMSOL simulations performed by the authors, but they could be presented in chapter 3.

Reply: We have taken into account the skin effect in the mathematical model of flow and transport in section 2 (see p. 10, lines 179-185 and p. 12, lines 224-225 ), and boundary and initial conditions have been presented in section 3.

13. l. 158: Is parameter "n" in eq. 6 explained in the text? I couldn't find it.

Reply: The parameter "n" in Eq. (15) has been explained in the text. See p. 15, lines 280-281. Thanks.

14. l. 180: "..the inner boundary condition inside the well..". It is unclear to which boundary the authors refer to. The well casing?

Reply: The inner boundary condition represents the boundary condition at r=rw, and we have explained it in the revised version. See p. 15, line 300.

15. l. 189: "During the rest phase, the solute flux from the borehole into the aquifer is zero,…". I don't agree with this statement. Given that the borehole has a finite dimension in the authors' simulations, there should be solute mass contained in the borehole at the end of the injection phase, and thus at the beginning of the rest phase. This solute should get flushed out of the borehole by regional groundwater flow.

Reply: Indeed, at the beginning of the drift phase, the injected solute should get flushed out of the borehole by regional groundwater flow, thus it could be a good idea that we set no BC in the borehole at this phase, as also suggested by the first reviewer. Thus, we have deleted this paragraph. See p. 13, lines 251-257.

16. Table 1: I couldn't find the skin radius rs in Table 1. Is there a reason not to list it?

Reply: The skin radius rs have been add into Table 1. Thanks for the careful check. See p. 39, line 755.

17. l. 228: "..progressively refined near the well." It remains unclear how fine the mesh size actually was near the well. Readers wanting to repeat the simulations will need to know.

Reply: We have added the mesh size near the well. See p. 17, line 330.

18. l. 239: Results: (1) The results of the COMSOL simulations are presented in an excessively large number of figures. The authors should carefully consider which figures are essential to providing new insights into the skin effect during PPTs (the main objective of their paper), and consider combining figures whenever possible. For example, Figs. 14 and 16 show PPT breakthrough curves affected by positive and negative skin effects. These two figures could easily be combined into a single figure. Other figures not immediately related to the main objective should be deleted or may be moved to a supplementary document. (2) The results are presented in qualitative fashion only, i.e., the reader can only visually compare the breakthrough curves between different simulations to judge tracer mass recovery. To improve this comparison the authors could, e.g., compute relative tracer mass recovered by the end of each PPT. This would allow for a more quantitative comparison.

Reply: (1) Figs. 10 and 12 have been combined into a single Fig.6 in this revised version, and Figs. 14 and 16 have been combined into a single Fig.9 in this revised version. In addition, previous figures (e.g. Fig.6, Fig.7 and Fig.8) have been moved into supplementary materials. See p. 53-56, Fig.6 and 7, p. 60-63. Fig.9 and 10. (2) To improve this comparison, we have computed the relative tracer mass recovered at the end of each SWPP test for different skin properties, and analyzed the impact of different hydraulic conductivities and skin thickness on the tracer mass recovered during the pumping phase. See section 4.4.

19. l. 250: ".. one can see that there is a stagnation point (Sp) located at the dividing streamline (Ds) as shown in Fig.4." This statement and figure are correct, but not new (see Monkmeyer and Netzer, 1993). Also, the term stagnation point is introduced without an explanation. What is the relevance of the stagnation point?

Reply: The Ds defines the capture zone boundary, and the Sp represents the uppermost location down-stream from the pumping well inside a capture zone. The region

beyond Sp in the down-stream direction cannot be captured by the pumping well. If tracer does not drift (with the regional flow only) beyond the stagnation point of the capture zone during the drift phase, it can be extracted from the aquifer. See p. 18, lines 358-366.

20. l. 270: the effect of resting time: The results of this section are a logical consequence of results from the previous section, where the effect of regional groundwater flow velocity are shown. Therefore, I would suggest to shorten this section and combine it with the previous.

Reply: The section about the effect of resting time have been moved into a supplementary as a reference.

21. l. 348: "Fig. 14 shows the effects of the skin thickness (positive skin) on BTCs during the pumping phase. One can see that the concentration gets higher at early stage with the increase of rs." This is not what is shown in Fig. 14, but rather the opposite (shown is highest early-time conc. for rs = 0). In fact, data plotted in Figs. 14 and 16 look identical. I suspect that the wrong set of data was plotted in Fig. 14. Furthermore, this is another example of two figures which could easily be combined into a single figure.

Reply: Thanks for pointing this out. We have corrected the error in previous Fig. 14 and combined previous Fig. 14 and Fig. 16 into a new Fig. 9 in this revised manuscript. See Fig.9.

22. l. 381: "Besides, the numerical model of SWPP test can be used to obtain unknown parameters: i.e., regional groundwater velocity, effective porosity, dispersivity, and biogeochemical reaction rates, by fitting to the observed BTCs." I find this conclusion unwarranted based on the merely qualitative results provided. First, inverse modeling is not a new element, this has been done before to assess parameters from PPTs (e.g., Gelhar and Collins, 1971, Schroth et al., 2001, Vandenbohede et al., 2008). But more importantly, the authors have not provided any data or sensitivity analysis for

this approach in their manuscript. It remains therefore unknown (and questionable) if such an inverse modeling approach will yield unique parameters sets with sufficient accuracy.

Reply: We agree with the reviewer that it is more important to analyze the effect of the skin with some experimental data. Unfortunately, we do not have such data at this stage. Further work will be conducted in the future, and will be reported elsewhere. In this manuscript, the main purpose is to offer a way to estimate unknown parameters: i.e., regional groundwater velocity, effective porosity, and dispersivity. In addition, we have also analyzed the impact of skin on the SWPP test, and analyze quantitatively the tracer mass recovered under the skin effect, and have conducted an error analysis for the non-skin model to interpret BTCs obtained from a model with a skin. The results indicate that the skin can produce considerable error for parameter estimations. See section 44, 4.5, and lines 587-592.

Please also note the supplement to this comment:
https://www.hydrol-earth-syst-sci-discuss.net/hess-2018-279/hess-2018-279-AC2-supplement.zip

---

## Author Comment (AC3) · 24 Oct 2018

Thank the reviewer very much for his/her careful check on the manusript. The point to point response can be found in the following. Please note that the referred page number and line number are referred to the marked version in the supplement files. The supplement files include: a marked version of the revised manuscript, a clean version of the revised manuscript, a supplementary material of the revised manuscript, and the response letter to all the referees' comments.

1. According to lines 114-116, the objective of this work is to study the impacts of skin effects on SWPP tests for estimating groundwater flow velocity. However, the article only shows that the skin effect has a significant influence on the shape of the BTC and

on the 2D distribution of concentration. It does not quantify the impacts of skin effects on SWPP tests "for estimating groundwater velocity". Knowing the impact of skin effects on the estimation of groundwater flow velocity (and of transport parameters) is of primary importance for practical applications. For example, in section 4.1 do the authors manage to estimate correctly the velocity in all cases? Ideally the authors should estimate groundwater flow velocity in the different skin configurations and compare it to the real one. At least, they should reformulate the objective of the paper and give an indication of the expected error on the identified groundwater flow velocity.

Reply: To quantify the impacts of skin effects on SWPP tests to estimating groundwater velocity, we have analyzed quantitatively the tracer mass recovered under the skin effect impact, and an error analysis has been conducted for the non-skin model to interpret BTCs obtained from the model with a skin. The results indicate that the skin can produce considerable error for parameter estimations. See section 4.4 and 4.5.

2. I am not sure to understand the reason why the authors show the impact of groundwater flow velocity on BTCs (section 4.1). Groundwater flow velocity affects the BTC and that is why SWPP tests can be used to estimate this parameter. It is interesting to better understand how groundwater flow velocity influences the SWPP test. However, it is not clear how this analysis is related to the objective of the paper (assessing the impacts of skin effects on SWPP tests). In the conclusions, the authors points out that groundwater flow velocity should be considered in order to design the experiment so that as much tracer as possible is recovered. This could be the reason that justifies section 4.1. Nevertheless, it is not explained how it is related to the objective of the paper.

Reply: One objective of this SWPP test is the determination of the unknown ambient groundwater velocity, and we need to know the characteristics and identifiable features of BTCs under different regional groundwater velocity scenarios. Therefore, types of BTCs should be analyzed for different regional groundwater velocity. To interpret this behavior further, we have introduced "dividing streamline" and "stagnation point" to

better understand how groundwater flow velocity influences the SWPP test. In the conclusions, we have pointed out that groundwater flow velocity should be considered in order to design the experiment so that as much tracer as possible can be recovered, and our purpose is to offer a proposal to estimate the groundwater velocity using the equation of Leap and Kaplan (1988). See p.18, lines 345-347 and p. 29, lines 586-590.

3. The same remark of point 2 can be done for the analysis of the impact of the duration of the rest phase, of porosity and dispersivity (sections 4.2, 4.3, 4.4).

Reply: The analysis of the impact of the duration of the rest phase, of porosity and dispersivity (sections 4.2, 4.3, 4.4) have been moved into a supplementary material as references.

4. Too many figures are presented: I suggest the authors to choose only the figures which are relevant to the objective of the article. In my opinion, figures 6, 7, 8, 9 are not necessary. Moreover, figures showing results with positive skin effects could be combined with figures showing results with negative skin effects.

Reply: Previous Figs.6-9 have been moved into a supplementary material as references. Previous Figs. 10 and 12 have been combined into a new Fig. 6 in the revised manuscript, and previous Figs. 14 and 16 have been combined into a new Fig. 9 in the revised manuscript. See p. 53-56, Fig.6 and 7, p. 60-63. Fig.9 and 10.

Technical corrections:

l.23 "the finite-element COMSOL Multiphysics": add "software".

Reply: Implemented, we have added it. See p. 2, line 27.

l.27 Dividing streamline: this sentence becomes clear only after having read the article. The authors should explain what they mean by "dividing streamline".

Reply: We have explained the terminology "dividing streamline" in the section 4.1, and the terminology has been deleted in the abstract. See p. 18, lines 360-363.

[Figure]

l.29-30 I think the sentence is not very clear. It could be reformulated as: "a smaller ratio between the hydraulic conductivity of the positive skin and that of the aquifer formation". Moreover, there is no need here to write the mathematical symbol\delta.

Reply: Implemented. See p. 2, lines 34-35.

Figure 1 The term "formation zone" is not very clear to me. Maybe it can be changed with "aquifer" or "aquifer formation". The caption should precise what are S1, S2, S3, S4. The coordinate axes are missing.

Reply: The term "formation zone" has been changed with "aquifer formation". See p. 10, line 177. S1, S2, S3, S4 have be explained in the caption precisely. The coordinate axes have been added in Fig. 1. See p. 37, line 722-723 and p. 43 Fig. 1.

l. 127 What is the "wellbore effect"? If this is important for the understanding of the paper it should be explained, otherwise it can be removed.

Reply: The term "wellbore effect" has been replaced with "wellbore storage". See p. 9, line 168.

l. 152 r was already defined at l.147. Reply: Implemented. See p. 10, line 192.

l. 159 I think it should be specified that H is head. Reply: H is the total head. See p. 15, lines 280-281.

l. 200 and l.70 Wang et al (2017) is missing in the references section Reply: Implemented. See p. 35, line 711-712.

l. 270 In the section title, symbol tres should be changed with "the duration of the rest phase"?

Reply: Implemented. See the supplementary material.

Figures 10 and 12: The symbol should be ndelta and not nsigma.

Reply: Implemented. See Fig.6.

Figures 11, 13 Why some of the flow lines are interrupted?

Reply: We have corrected them for 2D horizontal planes. See p. 56, Fig. 7 and p. 63, Fig. 10

l.331 "In contract to": I would rather say "In agreement with"

Reply: This paragraph containing this phrase "In contract to" has been deleted in the revised manuscript.

Figure 14 is probably wrong: concentration decreases with rs, differently from what is said at line 350.

Reply: We have corrected it. See p. 60, Fig. 9.

Figures 11, 13, 15 and 17: The skin zone is not very evident in the figures. Maybe it could be highlighted with a thicker line.

Reply: We have corrected it. See Fig. 7and Fig. 10.

Table 1: the skin thickness default value is missing.

Reply: We have corrected it. See Table 1.

Please also note the supplement to this comment:
https://www.hydrol-earth-syst-sci-discuss.net/hess-2018-279/hess-2018-279-AC3-supplement.zip